# SREBP-dependent lipidomic reprogramming as a broad-spectrum antiviral target

Shuofeng Yuan [1,2], Hin Chu[1,2], Jasper Fuk-Woo Chan[1,2,3,4,5], Zi-Wei Ye[2], Lei Wen[2], Bingpeng Yan[2], Pok-Man Lai[2], Kah-Meng Tee[2], Jingjing Huang[2], Dongdong Chen[2], Cun Li[2], Xiaoyu Zhao[2], Dong Yang[2], Man Chun Chiu[2], Cyril Yip[2], Vincent Kwok-Man Poon[2], Chris Chung-Sing Chan[2], Kong-Hung Sze[1,2], Jie Zhou[1,2], Ivy Hau-Yee Chan[6], Kin-Hang Kok [1,2,3], Kelvin Kai-Wang To[1,2,3,4], Richard Yi-Tsun Kao[1,2,3], Johnson Yiu-Nam Lau[3], Dong-Yan Jin [7], Stanley Perlman [8,9] & Kwok-Yung Yuen[1,2,3,4,5,10]

Viruses are obligate intracellular microbes that exploit the host metabolic machineries to meet their biosynthetic demands, making these host pathways potential therapeutic targets. Here, by exploring a lipid library, we show that AM580, a retinoid derivative and RAR-α agonist, is highly potent in interrupting the life cycle of diverse viruses including Middle East respiratory syndrome coronavirus and influenza A virus. Using click chemistry, the over-expressed sterol regulatory element binding protein (SREBP) is shown to interact with AM580, which accounts for its broad-spectrum antiviral activity. Mechanistic studies pinpoint multiple SREBP proteolytic processes and SREBP-regulated lipid biosynthesis pathways, including the downstream viral protein palmitoylation and double-membrane vesicles formation, that are indispensable for virus replication. Collectively, our study identifies a basic lipogenic transactivation event with broad relevance to human viral infections and represents SREBP as a potential target for the development of broad-spectrum antiviral strategies.

[1] State Key Laboratory of Emerging Infectious Diseases, Li Ka Shing Faculty of Medicine, The University of Hong Kong, Pokfulam, Hong Kong. [2] Department of Microbiology, Li Ka Shing Faculty of Medicine, The University of Hong Kong, Pokfulam, Hong Kong. [3] Carol Yu Centre for Infection, Li Ka Shing Faculty of Medicine, The University of Hong Kong, Pokfulam, Hong Kong. [4] Department of Clinical Microbiology and Infection Control, The University of Hong Kong-Shenzhen Hospital, Shenzhen 518000, China. [5] Hainan Medical University–The University of Hong Kong Joint Laboratory of Tropical Infectious Diseases, Hainan Medical University, Haikou 570100, China. [6] Department of Surgery, Li Ka Shing Faculty of Medicine, The University of Hong Kong, Pokfulam, Hong Kong. [7] School of Biomedical Sciences, Li Ka Shing Faculty of Medicine, The University of Hong Kong, Pokfulam, Hong Kong. [8] Department of Microbiology and Immunology, University of Iowa, Iowa City 52242, USA. [9] State Key Laboratory of Respiratory Disease, National Clinical Research Center for Respiratory Disease, Guangzhou Institute of Respiratory Health, The First Affiliated Hospital of Guangzhou Medical University, Guangzhou 510120, China. [10] The Collaborative Innovation Center for Diagnosis and Treatment of Infectious Diseases, Li Ka Shing Faculty of Medicine, The University of Hong Kong, Pokfulam, Hong Kong. These authors contributed equally: Shuofeng Yuan, Hin Chu, Jasper Fuk-Woo Chan. Correspondence and requests for materials should be addressed to K.-Y.Y. (email: kyyuen@hku.hk)

nfectious diseases account for ~20% of global mortality, and viruses are responsible for about one-third of these deaths[1]. In the past 20 years, emerging and re-emerging viruses such as severe acute respiratory syndrome (SARS) and Middle East respiratory syndrome (MERS) coronaviruses, avian influenza A (H5N1) and A(H7N9) viruses, pandemic 2009 influenza A virus (H1N1), Zika virus, and enteroviruses have posed significant global public health threats[2–8]. Rapid and effective control of these epidemics at their onset was often not possible due to the long time lag required for the development of specific antivirals or vaccines. Early empirical administration of a highly effective broad-spectrum antiviral would improve patients' outcome and facilitate the control of these epidemics if given before or soon after the exact pathogen is identified. Current strategies for the development of broad-spectrum antiviral agents mainly focus on two aspects, virus encoded targets, and host defense factors or cellular machineries that are exploited by viruses[9]. Successful examples of virus-targeting strategy include blockers of viral attachment and fusion[10,11], as well as inhibitors targeting viral enzymes, such as protease, polymerase, and neuraminidase, or internal structural proteins[12]. On the other aspect, type I interferons (IFNs) and IFN-induced proteins can be used to trigger the cellular machineries of host defense to suppress viral replication. Nevertheless, challenges of drug toxicity and emergence of resistant viral progenies remain to be addressed.

To fulfill the requirements of rapid and massive clonal replication, viruses must co-opt distinct programs to meet heightened metabolic demands. A key component in such reprogramming is the rapid up-regulation of lipid biosynthetic pathways, which can substantially impact on the viral replication process. Lipids have been recognized as structural elements of viral and cellular membranes. Viruses induce the formation of novel cytoplasmic membrane structures and compartments, in which viral genome replication and assembly occurs with perhaps shielding from host innate immune response. The involvement of lipids in the viral replication cycle is shared by enveloped and non-enveloped viruses, as well as both DNA and RNA viruses[13]. The correlation between virus infection and host lipid metabolism has been implicated in human cytomegalovirus (HCMV)[14]. Infection with HCMV markedly upregulated flux through much of the central carbon metabolism particularly in flux through the tricarboxylic acid cycle and its efflux to the fatty acid biosynthesis pathway.

Here, we demonstrate the essential role of lipid metabolic reprograming in MERS-CoV replication, an enveloped RNA virus highly divergent from HCMV. Thus, the modulation of cellular lipid metabolism to interfere with virus multiplication may be an appealing, broadly applicable approach for antiviral therapy. To this end, we carry out a pharmacological screening of a lipid library. AM580, a retinoid derivative and RAR-α agonist, demonstrates potent and broad-spectrum antiviral activities in vitro and in vivo. Using AM580 as a tool compound, we modify it by click chemistry and identify the host cell sterol regulatory element binding protein (SREBP) as the direct binding target of AM580. SREBPs are bHLH-zip transcription factors that have well-defined roles in the regulation of cellular lipid homeostasis. In mammals, there are two SREBP genes that express three SREBP proteins. SREBP1a and SREBP1c are produced via alternative transcriptional start sites on *Srebf1*, whereas the *Srebf2* gene encodes SREBP2. Canonical SREBP1c signaling preferentially drives expression of fatty acid biosynthesis genes whereas SREBP2 predominately transactivates genes involved in cholesterol biosynthesis, intracellular lipid movement and lipoprotein import[15]. Mechanistically, we find that AM580 blocks the interaction of SREBP1/2 proteins with the non-palindromic sterol regulatory elements (SREs) in the promoter/enhancer regions of multiple lipogenic genes, which inhibits their transcription and

thus reverses the virus-induced lipid hyper-biosynthesis. Collectively, our study identifies SREBP-mediated lipid biosynthesis with broad relevance to human viral infections and represents SREBP as an undescribed target for the development of broad-spectrum intervention strategies, especially for tackling novel viruses causing emerging infectious diseases.

## Results

**Virus infection reprogrammed host lipid metabolism**. To understand the host response to virus infection, we determined the transcriptomic profile of human bronchial epithelial Calu-3 cells infected with MERS-CoV. Gene Ontology (GO) enrichment of differentially expressed genes (DEGs) was performed to explore multiple aspects of virus–host interaction, including cellular processes, environmental information processing, genetic information processing, human diseases, organismal systems, and metabolism (Fig. 1a). Within the major category of metabolism, there is a combinational GO item with many global and overview maps that comprise sub-items such as carbon metabolism, fatty acid metabolism, biosynthesis of secondary metabolites, and biosynthesis of amino acids[16]. Notably, lipid metabolism was the top-ranking affected metabolic pathway, highlighting the importance of lipids among the metabolic changes triggered by MERS-CoV infection. In addition, the Kyoto Encyclopedia of Genes and Genomes (KEGG) enrichment analysis showed that steroid biosynthesis being the most highly enriched pathway, which indicated the markedly heightened lipid demands during MERS-CoV life cycle (Supplementary Figure 1a). To explore how MERS-CoV infection perturbed lipid metabolism, an untargeted lipidomic analysis was performed in MERS-CoV-infected Calu-3 cells. Lipid features significantly changed with $p < 0.05$ (Student's $t$-test) in MERS-CoV-infected cells were selected. For better demonstration of changed lipids profile trend after MERS-CoV infection, a Heatmap was constructed according to the identified lipid list (Supplementary Data 1). As shown in Fig. 1b, these changed lipid features include glycerophospholipid and fatty acids classes, indicating that MERS-CoV infection strongly perturbed lipid homeostasis. To map the landscape of metabolic-transcriptional alterations in the context of virus infection, we performed an integrated transcriptomic and lipidomic analysis to simultaneously map genes and lipid metabolites in different pathways. As shown in Fig. 1c, integrative network modeling specifically revealed that the glycerophospholipid metabolism pathway was most dramatically modulated by MERS-CoV infection. Collectively, we demonstrated that MERS-CoV infection triggered marked host lipid metabolic changes.

To investigate the importance of lipid metabolites in virus life cycle, we screened a lipid library using MERS-CoV and H1N1 virus infections. Colorimetric assays reflecting cell viability were performed to select compounds that inhibited the cytopathic effects (CPE) that develop upon virus infection (Fig. 1d). Screening conditions were optimized, in which at 0.1 MOI and 24 h post-MERS-CoV infection and at 0.01 MOI and 48 h post-H1N1 infection were chosen, respectively (Supplementary Figure 1b, c). A lipid metabolite with anti-inflammatory effects, 25-hydroxyvitamin D$_3$, protected Huh7 cells against MERS-CoV infection, while the aryl hydrocarbon receptor agonist FICZ and apoptosis regulatory messenger C16 Ceramide protected MDCK cells against H1N1 infection. Notably, AM580, a synthetic agonist, exhibited cell protection against both viruses and was selected for further investigations (Fig. 1d). Interestingly, (R)-methandamide, an agonist of cannabinoid receptor that functionally increased lipid accumulation in hepatocytes[17], facilitated virus replication at non-toxic concentrations, thus caused more

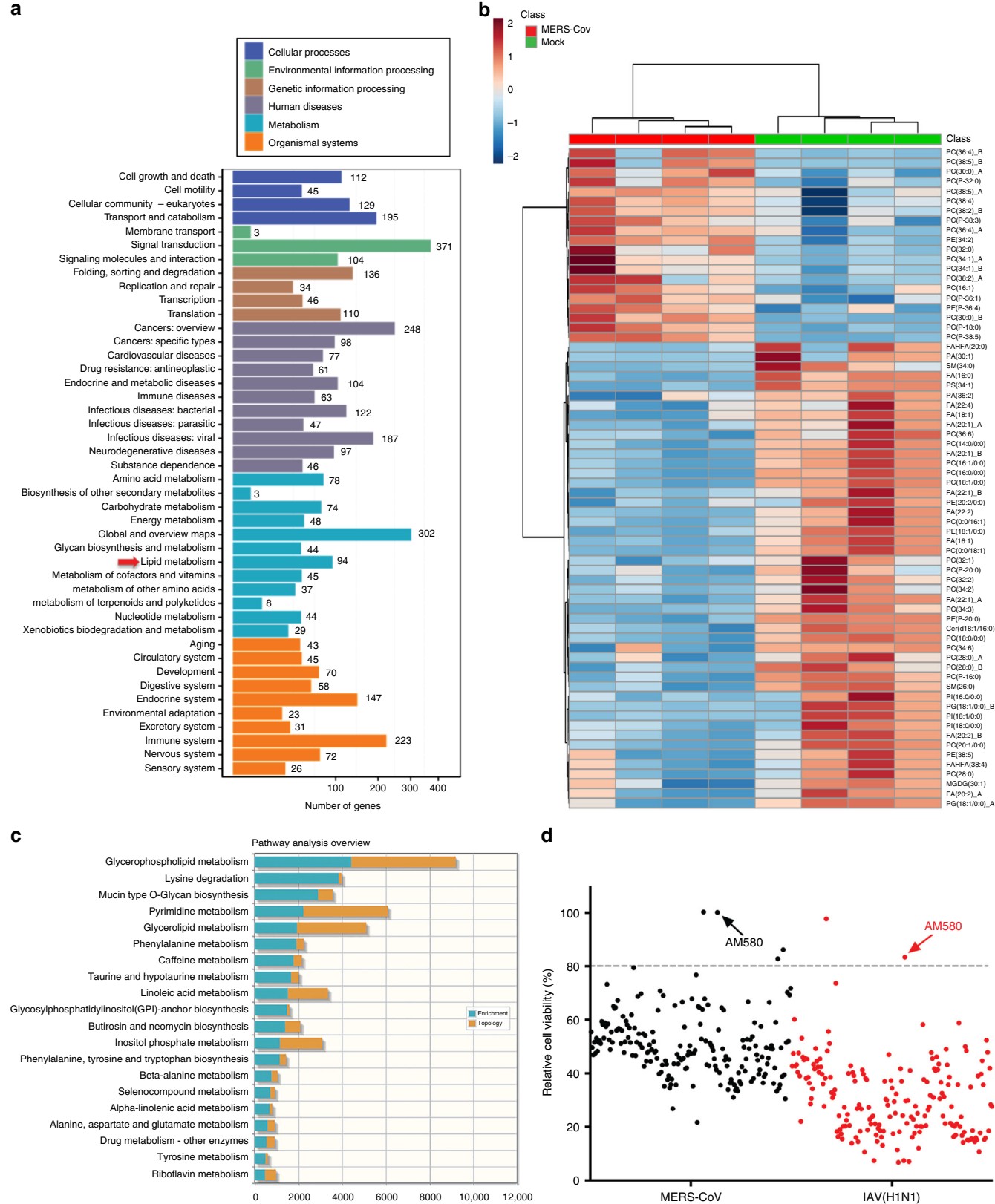

severe CPE than that of the virus control. These findings indicated that modulation of host lipid metabolism could significantly change the outcome of virus infection, thus corroborate with our hypothesis of modulating virus-induced lipid reprogramming for therapeutic intervention.

**AM580 exhibited broad-spectrum antiviral effects in vitro.** The cytotoxicity of AM580 was similar in different cell lines ($CC_{50}$: 100–200 μM; Supplementary Figure 2a). Using MERS-CoV infection as a model, we characterized the antiviral activity of AM580 in cell cultures. First, a multi-cycle virus growth assay was

**Fig. 1** Integrative transcriptomic–lipidomic analysis and lipid screening. Calu-3 cells were mock infected or infected with MERS-CoV at 2 MOI and incubated in DMEM medium. At 24 hpi, cells were harvested for transcriptomic (**a**) and lipidomic (**b**) analysis, respectively. **a** Gene Ontology (GO) analysis of differentially expressed genes (DEGs) in the MERS-CoV infected and non-infected Calu-3 cells. DEGs were classified under six categories as indicated. Red arrow indicates lipid metabolism as the top-ranking affected metabolism pathway. **b** Heatmap showing the lipidomic analysis of MERS-CoV-infected vs non-infected Calu-3 cells. Each rectangle represents a lipid colored by its normalized intensity scale from blue (decreased level) to red (increased level). The hierarchical clustering analysis was based on the identified lipid metabolites with significant changes in quantity. PC phosphatidylcholines, PA phosphatidic acid, PS phosphatidylserine, PC(P−) 1-(1z-alkenyl),2-acyl-phosphatidylcholine, PE phosphatidylethanolamine, PE(P−) 1-(1z-alkenyl),2-acyl-phosphatidylethanolamine, PG phosphatidylglycerols, PI phosphatidylinositol, MGDG monogalactosyldiacylglycerol, FA fatty acids, FAHFA fatty esters, SM sphingomyelin, Cer ceramides. **c** Integrated transcriptomic and lipidomic analysis. Both enrichment (blue) and topological (yellow) analysis are scored, indicating the glycerophospholipid metabolism as the most affected pathway after MERS-CoV infection. The analysis was performed by MetaboAnalyst 4.0. **d** Scatter plot showing the lipid library screening after MERS-CoV (black dots) or influenza A(H1N1) (red dots) infection. Cell viability with 0.1 MOI and 24 h post-MERS-CoV infection, and with 0.01 MOI and 48 h post-H1N1 infection were selected as the end-point of determination after drug treatment. Shown is the normalized result by setting mock-infection as 100%, which is averaged from three independent screenings

performed to plot the virus replication kinetics with or without AM580. Strikingly, AM580 treatment reduced viral titers in the cell supernatant by >3-$\log_{10}$ when compared with the dimethyl sulfoxide (DMSO) control, and the infectious plaque-forming units (PFUs) in the AM580 group remained at baseline levels during the whole time-course (Fig. 2a). Expression of MERS-CoV nucleoprotein (NP) was dramatically decreased upon AM580 addition, especially at 9 h post-infection (hpi) and 1 MOI (Fig. 2b). Flow cytometry showed that the percentage of MERS-CoV-infected cells after AM580 treatment decreased from 65% (DMSO) to 5.38% (AM580) at 24 hpi (upper panel, Fig. 2c). Furthermore, immunofluorescence staining for MERS-CoV NP suggested near-complete suppression of virus infection upon AM580 treatment (lower panel, Fig. 2c). AM580 also significantly reduced MERS-CoV replication in multiple cell types, including lung (A549 and Calu-3), kidney (Vero), and immune cells [THP-1 and human primary monocyte-derived macrophages (MDMs)] (Fig. 2d) as well as human primary small airway epithelial cells (HSAEC) (Fig. 2e). Moreover, AM580 suppressed virus-induced pro-inflammatory cytokine activation in Huh7 cells and MDMs (Supplementary Figure 3). Overall, AM580 showed potent anti-MERS-CoV activity in cell cultures with significant inhibition of virus replication, cell protection, and anti-inflammatory responses.

We previously established the human intestinal organoids (intestinoids) as an alternative route of virus transmission for MERS-CoV and an ideal tool for pharmacological evaluation[18]. Herein, we inoculated these intestinoids with 0.1 MOI of MERS-CoV and evaluated the effect of AM580. Our data demonstrated that AM580 treatment significantly ($p < 0.05$, one-way ANOVA) reduced MERS-CoV replication intra- and extra-cellularly (Fig. 2f, g). At 48 hpi, no PFU was detectable in AM580-treated intestinoid culture supernatants, representing a nearly 6-$\log_{10}$ reduction in titers when compared with DMSO-treated controls (Fig. 2f). The inhibition of MERS-CoV by AM580 was also evidenced by the markedly decreased expression of viral NP in the AM580-treated intestinoids when compared with the DMSO-treated intestinoids (Fig. 2h). Collectively, we demonstrated that AM580 robustly inhibited MERS-CoV replication in human intestinoids.

Our screening of a lipid library demonstrated the broad-spectrum antiviral potential of AM580 (Fig. 1d). Next, we investigated AM580's in vitro antiviral effect against other viruses especially the emerging or respiratory pathogens, including both RNA [SARS-CoV, H1N1 virus, enterovirus-A71 (EV-A71), and Zika virus (ZIKV)] and DNA [human adenovirus type 5 (AdV5)] viruses. Notably, AM580 inhibited the replication of all evaluated viruses at 50% inhibition concentration (IC$_{50}$) ranging from nanomolar to micromolar scales in a concentration-dependent

manner (Fig. 2i). The selectivity index of AM580 was remarkable for most of the tested viruses, in particular for MERS-CoV (507), SARS-CoV (114), and influenza A(H1N1)pdm09 virus (159), suggesting the potential for safe usage of AM580 or its clinically available analogs in therapeutic settings for a broad spectrum of viral pathogens (Supplementary Figure 2c).

**AM580 exhibited broad-spectrum antiviral effects in vivo**. To evaluate the in vivo antiviral activity of AM580, we first examined whether the drug conferred protection against lethal challenge with MERS-CoV in human DPP4 (hDPP4)-transgenic mice[18]. Previous pharmacokinetics study revealed that AM580 has a relatively large apparent volume of distribution (1.1–1.5/kg) and small clearance (8.8–9.7 ml/min/kg), with pharmacokinetic behavior linear within the dose range from 2 to 10 mg/kg after intraperitoneal injection[19]. Intraperitoneal (i.p.) inoculation of the maximal amount of AM580 soluble in PBS (12.5 mg/kg) for 7 days resulted in no signs of toxicity (Supplementary Figure 2b). In mice challenged with 50 PFU of MERS-CoV, all 20/20 mice (100%) survived after they received 3 days of i.p. injection of AM580, whereas 14/20 DMSO-treated mice died (survival rate 30%; Fig. 3a). Mice in the AM580-treated group exhibited significantly less body weight loss ($p < 0.05$, Student's $t$-test) than that of the DMSO-treated group on days 4 and 5 post-infection (Fig. 3b), and with lower lung tissue virus titers and viral loads ($p < 0.01$, Student's $t$-test) at days 2 and 4 (Fig. 3c; Supplementary Figure 7e). On day 4 post-challenge, the viral RNA load in brain tissues of the AM580-treated mice was undetectable and was 4-$\log_{10}$ lower than that of the DMSO-treated mice (Supplementary Figure 7e). Histopathologic examination showed that alveolar damage and interstitial inflammatory infiltration in the lung tissues of the AM580-treated mice were substantially diminished compared to the control mice (Fig. 3d). Collectively, these results demonstrated that AM580 effectively protected hDPP4-transgenic mice from lethal MERS-CoV challenge by inhibiting MERS-CoV replication and virus-associated pneumonia and encephalitis in vivo.

In parallel, the antiviral effects of AM580 against the highly pathogenic avian influenza A virus (H7N9) were evaluated in Balb/c mice[20]. In mice challenged with 100 PFU of H7N9, intranasal AM580 treatment resulted in significantly higher survival rates (6/10, 60%; vs 0/10 in DMSO-treated controls, 0%; $p < 0.01$, log-rank test) (Fig. 3e). AM580-treated mice showed less body weight loss than that of DMSO-treated group from day 3 to day 7 post-infection (Fig. 3f). The mean viral RNA load in the lung tissues of AM580-treated mice was significantly ($p < 0.01$, Student's $t$-test) lower than that of control mice (Fig. 3g; Supplementary Figure 7f). Histopathological examination

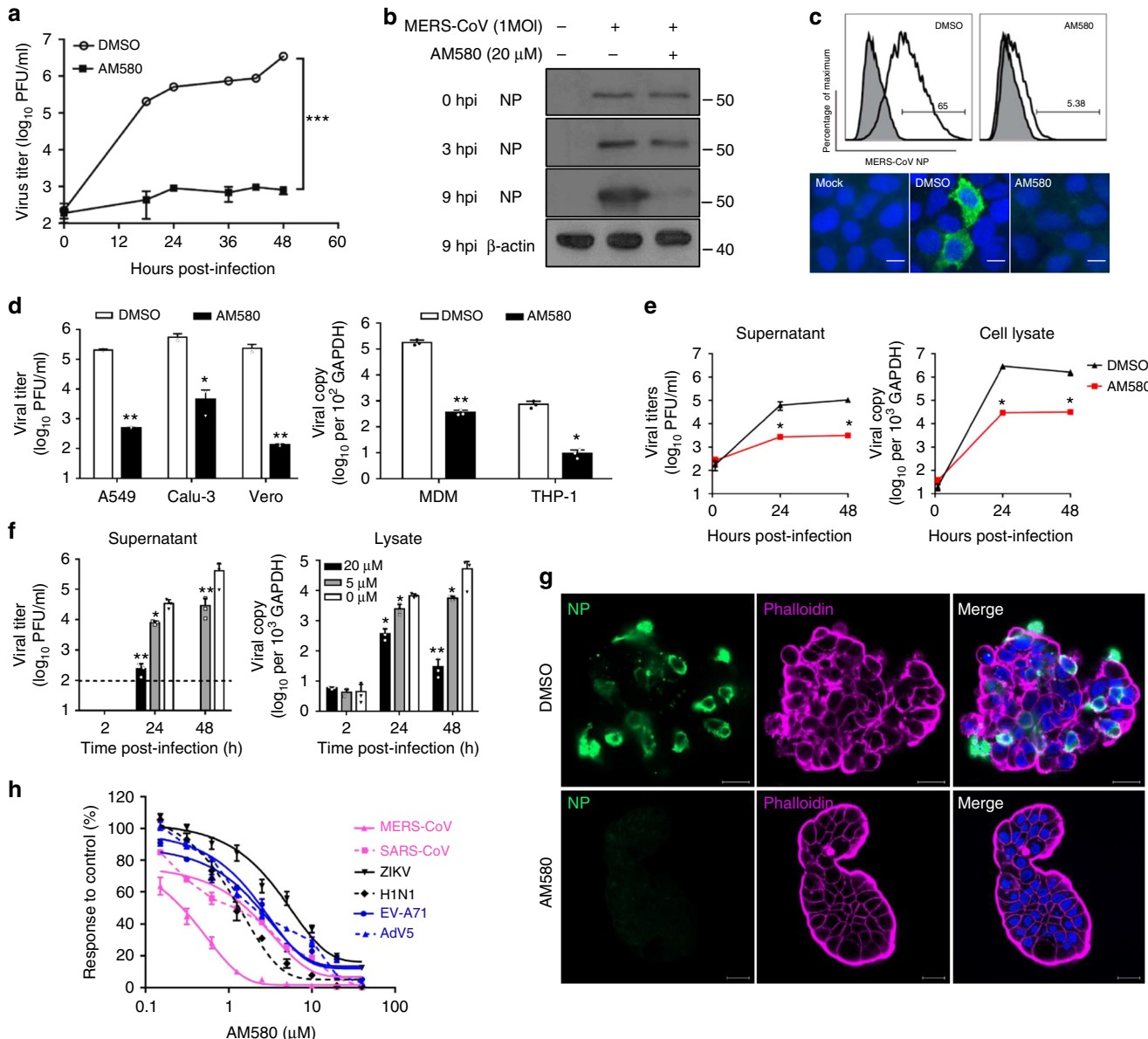

**Fig. 2** In vitro antiviral activity of AM580. **a** Multi-cycle MERS-CoV growth assay in the presence or absence of AM580. Huh7 cells were infected with MERS-CoV (0.01 MOI). Viral titers in cell culture supernatants were quantified by plaque assay at different time points. Differences between DMSO (open circle) and AM580 (20 μM, black square) groups were analyzed by Student's $t$-test. **b** Western blot showed reduced MERS-CoV NP production after AM580 treatment. Huh7 cells were infected with 1 MOI MERS-CoV. **c** Upper panel: MERS-CoV-NP-positive cells quantitated by flow cytometry. Lower panel: immunofluorescence staining of MERS-CoV-NP antigen (green) and cell nucleus (blue). Scale bar:10 μm. **d** AM580 reduced MERS-CoV replication in cell culture supernatants of A549 (0.1 MOI), Calu-3 (0.1 MOI), and Vero (0.01 MOI) cells at 24 hpi. AM580 reduced MERS-CoV replication in cell lysates of monocyte-derived macrophage cells (MDM, 1 MOI) and THP-1 (0.1 MOI) at 24 hpi. Differences between DMSO and AM580 groups were analyzed by Student's $t$-test. **e** AM580 inhibited MERS-CoV replication in human primary small airway epithelial cells (HSAEC) that were infected by 1 MOI MERS-CoV and treated with (red square) or without AM580 (black triangle). Supernatant and cell lysate were collected at the indicated time points and titrated by plaque assay and RT-qPCR assay, respectively. **f, g** AM580 showed anti-MERS-CoV activity in human intestinal organoid (intestinoid). **f** One-way ANOVA was used for comparison of the AM580 treated with the DMSO treated. **g** Representative images of intestinoids, after immunofluorescence staining for MERS-CoV NP (green), DAPI, and Phalloidin (purple), were 3D-imaged with a confocal microscope. Scale bar: 20 μm. **h** AM580 showed broad-spectrum antiviral effects against six different viruses as indicated. Plaque reduction assays were performed to evaluate antiviral activity of AM580 in MERS-CoV (Huh7 cells, magenta triangle) and SARS-CoV (Huh7 cells, magenta rectangle), ZIKV (Vero cells, black triangle), H1N1 virus (MDCK cells, black diamond), EV-A71 (RD cells, blue dot). $TCID_{50}$ assays were used for AdV5 titration (HEp-2 cells, blue triangle). Shown are the PFU or $TCID_{50}$ of indicated concentrations relative to controls in the absence of compound (%). The experiments were performed in triplicate and replicated twice. The results are shown as mean ± s.d. \*\*\*$p < 0.001$, \*\*$p < 0.01$, \*$p < 0.05$

revealed that AM580 treatment ameliorated virus-associated pulmonary inflammatory infiltration and bronchopneumonia (Fig. 3h). Taken together, our results demonstrated the in vivo protective effects of AM580 against lethal H7N9 infection.

**SREBPs are essential for virus replication**. The extraordinary potency and broad antiviral spectrum of AM580 hinted that its cellular target should be a vulnerable and upstream effector. Using AM580 as a tool compound, we try to decipher the key

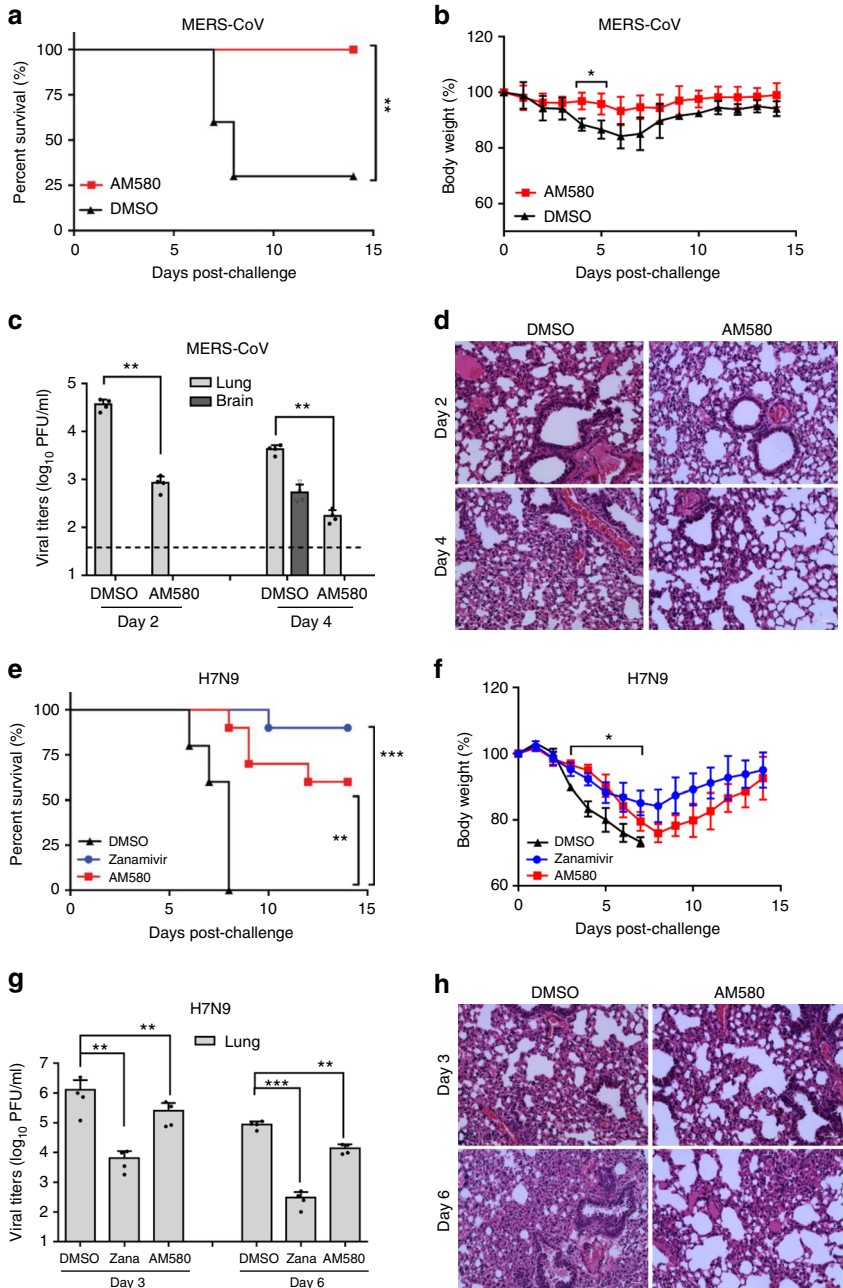

**Fig. 3** In vivo antiviral activity of AM580. **a** DDP4 transgenic mice were treated by intraperitoneal inoculation of AM580 (red square) or 0.1% DMSO (placebo control, black triangle) for 3 days starting 6 h post-challenge with 50PFU of MERS-CoV. Survivals and clinical disease were monitored for 14 days or until death. Differences in survival rates between groups were compared using Log-rank (Mantel–Cox) tests. **b** Daily body weights of surviving mice. Student's *t*-test was used to compare different groups on each day post-infection. **c** Lung and brain tissues were collected for detection of viral yields using both plaque assay and RT-qPCR assays (Supplementary Figure 7). Difference were compared with DMSO-treated groups using Student's *t*-test. **d** Representative histologic sections of lung tissues from the indicated groups with hematoxylin and eosin (H&E) staining. Greater alveolar damage and interstitial inflammatory infiltration were present in the DMSO group. **e–h** Balb/c mice were treated with intranasal AM580 (red square), zanamivir (positive control, blue dots), or 0.1% DMSO (negative control, black triangle) for 3 days starting 6 h post-challenge with 100 PFU of influenza A (H7N9) virus. Shown are survival rate (**e**), mean body weight (**f**), lung viral load (**g**), and representative lung sections stained by H&E (**h**). The same statistical analyses were performed as described in **a–d**. Results are presented as mean values ± s.d. ***$p < 0.001$,**$p < 0.01$, *$p < 0.05$. Scale bar: 20 μm

components of lipid metabolism that exhibited broad relevance to human viral infections. First, we used an immunofluorescence test to visualize the distribution patterns of cellular lipid droplets (LDs) and cholesterol in the presence or absence of AM580 in MERS-CoV-infected Huh7 cells, which are liver cells highly active in lipid metabolism. Infection by MERS-CoV markedly enhanced the accumulation of LDs and cholesterol, whereas addition of

AM580 significantly reduced their accumulation (Fig. 4a). To validate this observation, mRNA expression of lipogenic genes in the lipid biosynthesis pathways were investigated. Significant decreases in mRNA expression were detected in 13/17 (76.5%) genes involved in fatty acid biosynthesis (upper panel, Fig. 4b) and 10/12 (83.3%) genes in cholesterol synthesis pathways (lower panel, Fig. 4b) in AM580-treated infected cells, when compared

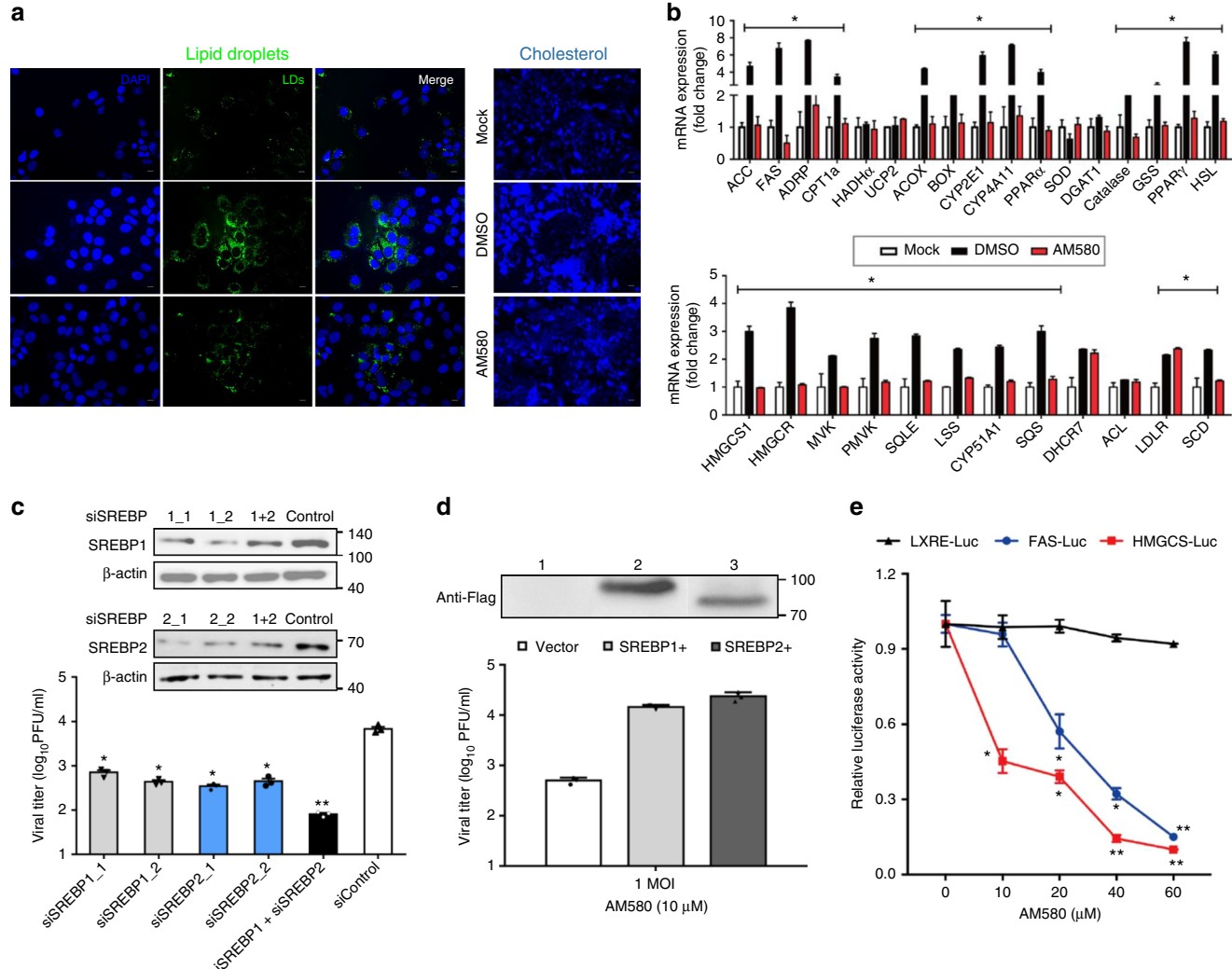

**Fig. 4** SREBPs were essential for MERS-CoV replication. **a** AM580 decreased cellular lipid droplets (LDs) and cholesterol levels. Huh7 cells were infected with MERS-CoV at 0.01 MOI for 24 h, in the presence of 0.1% DMSO or 20 μM AM580 or mock-infected. Cells were fixed and stained with DAPI (blue) and BODIPY 493/503 lipid probe (green) for LDs detection, or with filipin (blue) for intracellular cholesterol visualization. Scale bar: 20 μm. **b** The expression of multiple genes in fatty acid (upper) and cholesterol (lower) synthesis pathways were analyzed by RT-qPCR, respectively. * indicates $p < 0.05$ comparing DMSO and AM580 by one-way ANOVA. **c** siRNA knockdown of either SREBP1 (two distinct siRNA 1_1 and 1_2) or 2 (siRNA 2_1 and 2_2) or both SREBP1 and 2 (siRNA 1_2 and 2_2) reduced MERS-CoV replication. The knockdown efficiency and specificity was evaluated with western blot. siRNA-treated cells (100 nM siRNA for single knockdown and 50 nM each for double knockdown for 48 h) were infected with MERS-CoV (0.001 MOI and 24 hpi). Viral titer in the cell culture supernatant was evaluated with plaque assay. One-way ANOVA was used for comparison with the control siRNA pre-treated group. **d** Hyper-expression of nuclear form SREBP1 and SREBP2 by transfection to Huh7 cells for 24 h, respectively, followed by MERS-CoV infection and (1 MOI, 12 hpi) AM580 treatment as indicated. Overexpression of n-SREBP1 and n-SREBP2 were analyzed by western blot using anti-flag tag antibodies. Differences in viral titer were compared with the vector-transfected control and analyzed using Student's $t$-test. **e** AM580 inhibited transactivation of lipogenic genes such as FAS (blue dots) and HMGCS (red square) but not LXRE (black triangle). Huh7 cells transfected with the indicated reporter plasmid were treated with serial-dilutions of AM580 for 24 h. For virus infection assays, AM580 was added after virus absorption; for luciferase assay, AM580 was added 6 h after plasmid transfection. Student's $t$-test was used to compare the AM580-treated with DMSO-treated groups. The experiments were performed in triplicate and replicated twice. The results are shown as mean ± s.d. **$p < 0.01$, *$p < 0.05$

with those of the DMSO-treated infected controls. Moreover, the profound increase of the major lipogenic enzymes, including acetyl-CoA carboxylase (ACC), fatty acid synthase (FAS), and HMG-CoA synthase (HMGCS) associated with MERS-CoV infection indicated that the virus rapidly upregulated lipid biosynthesis, while AM580 might antagonize these changes, thereby reducing virus replication.

Searching of the upstream proteins regulating the host lipogenic pathway showed that SREBPs are the major factors that control lipid biosynthesis through transactivation of genes encoding the lipogenic enzyme[21]. To evaluate the role of SREBPs

on MERS-CoV replication, we compared the growth of MERS-CoV in wild-type control cells (mock) and SREBP gene silencing or hyper-expression cells. Notably, transfection of SREBP1- or SREBP2-targeted siRNAs diminished the precursor SREBPs (pre-SREBPs) production, which significantly ($p < 0.05$, one-way ANOVA) reduced MERS-CoV replication (Fig. 4c). Though the expression level of SREBP1 and SREBP2 in double knockdown is not more reduced than that of the individual knockdown because only 50% the amount of siRNA for each SREBP was applied, the double knockdown of both SREBPs resulted in about 2 $\log_{10}$ ($p < 0.01$, one-way ANOVA) decrease of infectious virus particle when

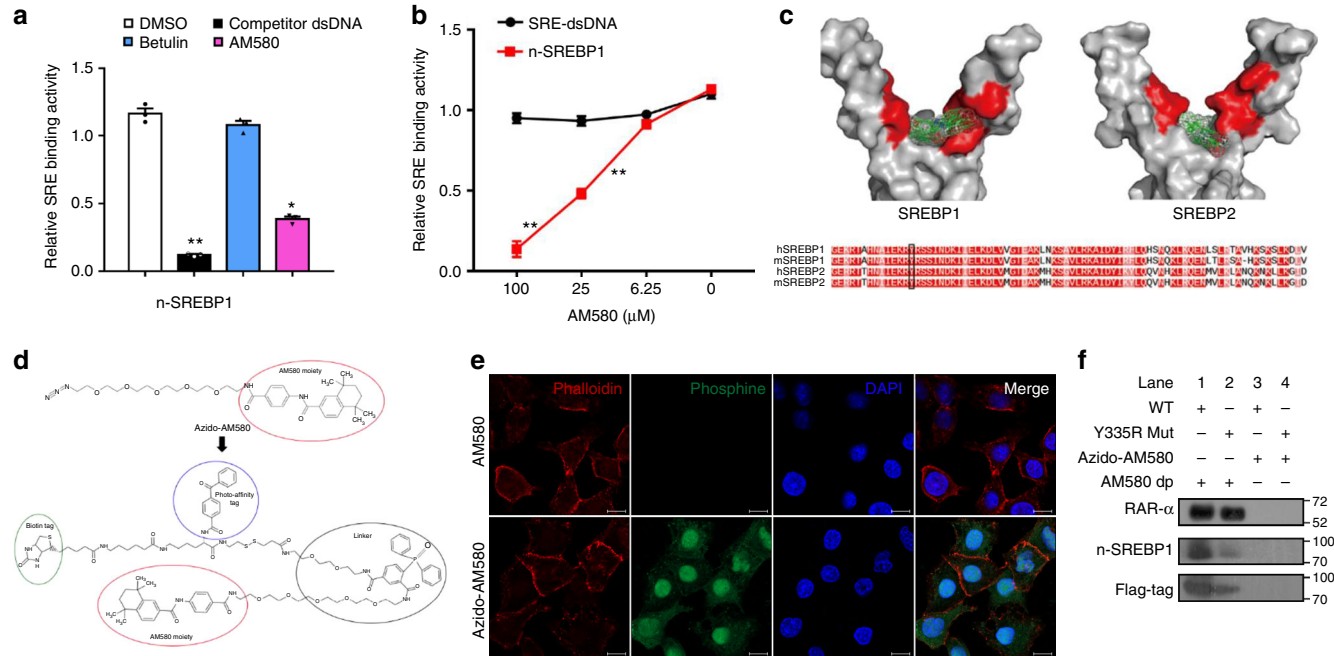

**Fig. 5** AM580 interacted with n-SREBP to block lipogenic transactivation. **a** AM580 blocked n-SREBP1 and SRE binding. DNA-binding activity of nuclear-extracted SREBP1 (n-SREBP1) to double-strand DNA (a mimic of SRE) immobilized onto the wells of microtiter plates. Betulin, negative control; competitor dsDNA, positive control. One-way ANOVA was used for comparison with the DMSO-treated group. **b** AM580 bound with n-SREBP1 instead of SRE. Black line indicates AM580 was added and incubated in SRE-dsDNA-immobilized wells, washed before addition of n-SREBP1; red line with square indicates AM580 was pre-incubated with n-SREBP1 before adding to SRE-dsDNA-immobilized wells. Students' $t$-test was done between groups with same concentrations of AM580 but different treatments. The experiments were performed in triplicate and replicated twice. The results are shown as mean ± s. d. $**p < 0.01$, $*p < 0.05$. **c** AM580 was predicted to occupy the SRE-recognition sites of both SREBP1 and 2. Shown is the 3D molecular docking analysis. Potential interaction surfaces on SREBPs (red) are shown, while AM580 (green) is displayed in stick and mesh representation. Partial sequence alignment of the DNA-binding domains of human and mouse SREBP1 and 2 is shown. Tyr335, the key residue for AM580 binding is highlighted with a box. **d** Structure of an AM580-derived probe (AM580dp) showing the locations of designated groups with specific functionalities. Azido-AM580 was synthesized through the reaction between azido-PEG5-amine and AM580. AM580dp was further synthesized through the addition of tri-functional crosslinker with azido-AM580. **e** Cellular distribution of azido-AM580. AM580 was used as a negative control due to the lack of phosphine-specific azido group. Scale bar:10 μm. **f** Tyr335 was critical for AM580 and n-SREBP1 interaction. AM580dp was immobilized on streptavidin beads and incubated with the cell lysate that were transfected with either WT or Y335R mutant constructs. After pull-down, western blot was employed to detect n-SREBP1 using both anti-n-SREBP1 and anti-Flag antibodies. Overexpression of RAR-α protein was used as a positive control for the pull-down capacity of AM580dp, while azido-AM580 was used as a negative control to exclude non-specific binding

compared with the 1 $\log_{10}$ reduction of individual knockdown, which indicated a potential synergistic effects when both activities of SREBP1 and SREBP2 were inhibited. Together, this result indicated that SREBPs were essential for MERS-CoV replication. Transactivation of lipid biosynthesis genes requires cleavage of pre-SREBPs to release their N-terminal nuclear forms (n-SREBPs), a process which is regulated by sterols[22]. Overexpression of either n-SREBP1 or n-SREBP2, however, diminished the antiviral potency of AM580 by about 1.5-$\log_{10}$ upon MERS-CoV infection (Fig. 4d). Together, these results indicated a pro-viral role of n-SREBPs in MERS-CoV replication and an inhibitory role of AM580 on n-SREBPs' activity.

To further confirm if the functionality of SREBP1 and/or SREBP2 are affected by AM580, reporter gene assays reflecting SREBPs-dependent transcriptional activation were performed. As shown in Fig. 4e, lipogenic enzymes such as HMGCS and FAS were blocked at the transcriptional level. However, the Liver X Receptor response element (LXRE), which is required for the activation of SREBP[23], was not affected. These results indicated that AM580 might specifically disrupt the transactivation of lipogenic enzymes mediated by n-SREBPs.

**AM580 blocked n-SREBPs binding to SREs.** FAS and HMGCS are two lipogenic enzymes that belong to separate lipid

biosynthesis pathways. Inhibition of both genes transactivation, as shown in Fig. 4e, led us to suspect that AM580 functions to interrupt the interaction of n-SREBPs with the SREs that are conserved in lipogenic promoters including FAS, ACC, HMGCS, etc. To this end, SREBP1/2 transcription factor assays were performed, in which specific double stranded DNA (dsDNA) sequences containing the n-SREBP1- or n-SREBP2-binding SREs were immobilized in the solid phase. Binding intensity of nuclear-extracted n-SREBP1 or n-SREBP2 was then detected in the presence or absence of inhibitors. Notably, AM580 inhibited the binding of both n-SREBP1 (Fig. 5a) and n-SREBP2 (Supplementary Figure 5a) with their corresponding SREs, whereas the compounds FICZ and 25-hydroxyvitamin $D_3$ did not (Supplementary Figure 7m). To determine whether AM580 targets n-SREBPs or SREs, we pre-incubated AM580 with either immobilized SRE-dsDNA before adding n-SREBPs or with n-SREBPs before binding with SRE-dsDNA. Taking n-SREBP1 as an example, AM580 was found to bind with n-SREBP1 and not SRE-dsDNA, and inhibited the SRE-dsDNA binding activity of n-SREBP1 in a dose-dependent fashion (Fig. 5b).

Next, to determine the interacting residue(s) on n-SREBPs that are responsible for AM580 binding, we first performed molecular docking analysis using the published crystal structures of SREBP1 and SREBP2 (refs. [24,25]). The V-shape DNA-binding domain of

n-SREBP1 and 2 shows structurally similarity and binds AM580 as a homodimer (Fig. 5c). Taking n-SREBP1 as an example, AM580 binds to the E-box site through four amino acids (His328, Glu332, Tyr335, and Arg336) that are highly conserved among helix–loop–helix proteins (Supplementary Figure 5b). The major interaction was predicted to be between AM580 and residue Tyr335 by a hydrogen bond. Importantly, the Tyr335 that determines SRE recognition[24] is completely conserved between human and mouse SREBPs (lower panel, Fig. 5c), which may explain the conservation of the broad spectrum antiviral property of AM580 in human cells and in mouse models.

To explore if AM580 inhibits the DNA-binding activity of SREBPs by physically blocking SRE recognition (i.e. via Tyr335), we introduced an Y335R mutation into n-SREBP1 and compared its AM580 binding capacity with that of the wild-type protein. An AM580-derived probe with biotin and photo-affinity tags (AM580dp) was synthesized to facilitate the evaluation of its binding characteristics. The AM580dp was made by introducing a linker arm containing an azido end to the carboxylic acid group on AM580, yielding the azido-AM580, which was designed for further addition of a tri-functional linker with UV photo-affinity and biotin tags with specific probing functionalities (Fig. 5d). Like AM580, azido-AM580 inhibited MERS-CoV replication in Huh7 and Vero cells (Supplementary Figure 5c). Next, using an azido-reactive green fluorescent dye for localization, azido-AM580 was found largely in the host cell nucleus, which corroborated our hypothesis that AM580 may target a lipogenic transactivation event (Fig. 5e). To capture the binding target, AM580dp was immobilized on streptavidin-conjugated agarose by its biotin group and incubated with transfected cell lysates expressing WT or Y335R n-SREBP1. RAR-α, a known AM580 receptor, was co-transfected as a control. After ultraviolet irradiation to activate the crosslinking photo-affinity tag in AM580dp, the protein-AM580dp complex was fixed and isolated. As shown in Fig. 5f, almost equal amounts of RAR-α were precipitated by AM580dp in both WT or mutant Y335R n-SREBP1 groups, indicating that AM580dp was biologically functional. However, significantly more WT n-SREBP1 was detected than Y335R n-SREBP1, suggesting a stronger binding affinity between AM580 and WT n-SREBP1 than that of Y335R n-SREBP1. In addition, SRE binding activity of n-SREBP1 was significantly ($p < 0.01$, Student's $t$-test) diminished when Tyr335 was substituted with arginine, which highlighted the crucial role of Tyr335 in SRE recognition (Supplementary Figure 5d). Taken together, we concluded that AM580 disrupted n-SREBP1 and SRE binding, specifically via impairing the SRE-recognition functionality of n-SREBP1.

**Block of SREBPs-dependent pathways reduced viral fitness.** Disruption of n-SREBP and SRE interaction leads to failure of lipogenic transactivation. To elucidate these downstream consequences, we investigated one of the SREBP-mediated pathways, fatty acid synthesis. First, we explored the ability of an end-product of the de novo fatty acid biosynthesis pathway, sodium palmitate, to reverse the antiviral activity of AM580. Notably, both C75 (a FAS inhibitor specific for fatty acid biosynthesis pathway) and AM580 showed anti-MERS-CoV activity (Fig. 6a). Addition of sodium palmitate did not affect the virus yield in MERS-CoV-infected cells treated with DMSO, but increased the virus yield ($p < 0.05$, one-way ANOVA) in MERS-CoV-infected cells treated with either AM580 or C75 (Fig. 6a). This finding suggested that MERS-CoV replication hijacked host fatty acid synthesis, which can be inhibited through shut-down of lipogenic transactivation while rescued by exogenous palmitate. Next, we explored whether fatty acid synthesis was critically involved in the replication of other AM580-inhibited viruses. To this end,

replication rescue assays using H1N1 (negative-strand RNA virus representative), EV-A71 (non-enveloped RNA virus representative), and AdV5 (DNA virus representative) were performed (Fig. 6b, d). Indeed, significant ($p < 0.05$, Student's $t$-test) extents of rescue were achieved for these viruses and especially H1N1 virus ($p < 0.01$, Student's $t$-test) with addition of sodium palmitate.

Positive-sense RNA viruses are known to replicate their genomes on intracellular membranes. For MERS-CoV, double-membrane vesicles (DMVs) provide the anchoring scaffold for viral replication/transcription complexes, which might be disrupted by the blockade of fatty acid synthesis. Using electron microscopy, perinuclear DMV clusters were readily detectable in MERS-CoV-infected cells (left panel, Fig. 6e). In contrast, almost no DMVs were visualized after treatment by AM580 (right panel, Fig. 6e). Since co-expression of nsp3 and nsp4 of MERS-CoV was sufficient to induce the DMV formation[26], we used AM580 to treat the Huh7 cells co-transfected with nsp3 (~209 kDa) and nsp4 (~57 kDa) for 24 h. As shown in Fig. 6f, no significant changes were detected in the expression levels of both viral proteins. The finding indicated that the reduced DMV formation was caused by the inhibition of lipogenesis directly and not by the decreased viral replication indirectly.

Negative-sense RNA viruses, such as influenza A viruses, utilize a different mechanism of genome replication and transcription that is independent of intracellular replicative organelles. Palmitoylation, a downstream consequence of fatty acid synthesis, is a post-translational modification that modulates protein function and protein localization. In the context of influenza A viruses, the best characterized palmitoylated protein is the surface glycoprotein hemagglutinin (HA)[27]. Therefore, we explored whether blockade of fatty acid synthesis would impede influenza HA palmitoylation and inhibit H1N1 replication. HA-overexpressed A549 cells were cultured with AM580 or vehicle (DMSO) or the positive control inhibitor 2-BP, which is specific against protein palmitoylation[28]. Palmitoylated HA protein was purified via resin-assisted capture. Strikingly, reduced levels of palmitoylated HA were observed with the addition of 2-BP (56%) and AM580 (69%) when compared with DMSO (100%) controls (Fig. 6f). Moreover, 2-BP also showed inhibition against MERS-CoV and H1N1 replication in a dose-dependent manner (Supplementary Figure 4b). To examine the specificity, oleic acid, a downstream metabolite of palmitate was also used in the complementation assays. The result showed that additional oleic acid (100 μM) could not rescue the influenza A H1N1 virus replication but indeed complement the viral growth of MERS-CoV, ZIKV, and AdV5 (Fig. 6a, c, d). This finding suggested that the palmitoylation of the viral HA was the main target of influenza A H1N1 virus inhibition by AM580. Overall, using DMV formation and viral protein palmitoylation as two important downstream consequences of SREBPs-dependent pathways, we demonstrated that the suppression of SREBPs-dependent lipogenic transactivation reduced viral propagation fitness by intervening the downstream fatty acid biosynthetic pathway.

## Discussion

Viruses rely on the metabolic network of their cellular hosts to provide energy and building blocks for replication. Previous studies on the viral perturbation of host lipid metabolism have not been able to identify the underlying mechanism and host factors targeted by the invading virus. The remarkable in vitro and in vivo antiviral efficacy of the tool compound AM580, through the inhibition of SREBP-related pathways, confirms that up-regulation of host lipid biosynthesis is crucial in multiple

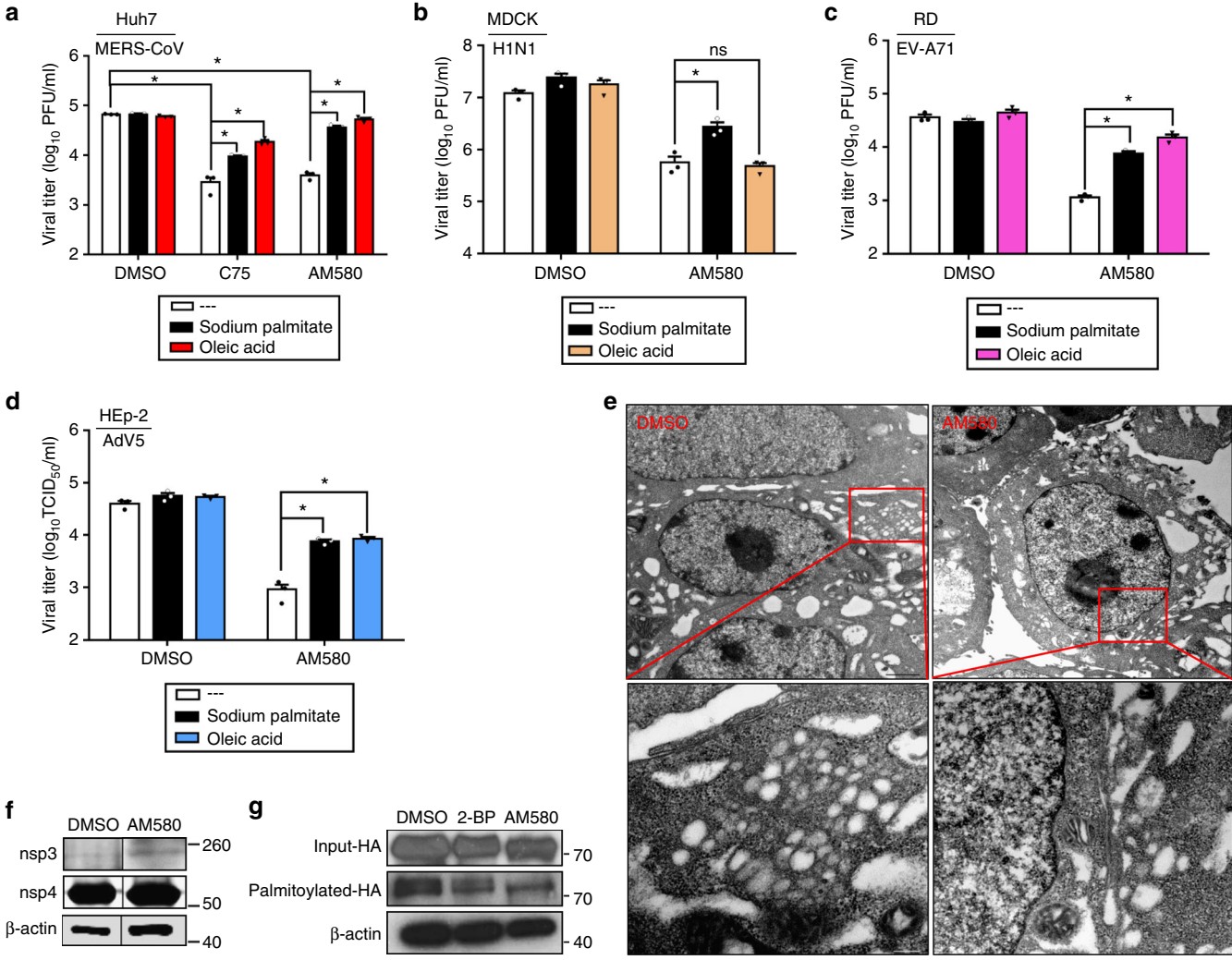

**Fig. 6** Suppressed SREBP-dependent pathways reduced virus replication. **a** Inhibition the fatty acid synthesis reduced virus replication, while exogenous palmitate or oleic acid rescued MERS-CoV replication. Huh7 cells infected with MERS-CoV (0.01 MOI) were treated with DMSO, or C75 (FAS inhibitor), or AM580 in the absence (white bars) or presence of supplemental exogenous palmitate (black bars) or oleic acid (red bars). Viral titers in culture supernatants after 24 hpi are shown. Differences between groups were analyzed by one-way ANOVA test. **b–d** Virus rescue assays were performed for H1N1 virus (0.001 MOI), EV-A71 (0.001 MOI), and human AdV5 (100TCID$_{50}$) in different cell lines as described in **a**. Viral titer for different viruses were analyzed by Students' $t$-test. *$p < 0.05$, **$p < 0.01$. The experiments were performed in triplicate and replicated twice for confirmation. The results are shown as mean ± s.d. **e** AM580 inhibited DMVs formation. Vero cells were infected with 3 MOI of MERS-CoV and treated with DMSO or AM580 for another 12 h before processing for the staining before transmission electron microscopy. Virus-infected cells without treatment showed perinuclear clusters of DMVs (red box) and the lack of DMVs production upon AM580 treatment. Representative electron microscopy images were selected from a pool of over 30 images captured in two separate experiments. Scale bar of upper panel: 1 μm, lower panel: 200 nm. **f** AM580 did not affect viral protein expression of nsp3 and nsp4. Huh7 cells were co-transfected with MERS-CoV nsp3 and nsp4 with flag tag, and treated with AM580 at 6 h post-transfection for another 24 h. Protein expression level was evaluated by western blot using β-actin as an internal control. **g** AM580 reduced viral protein palmitoylation. A549 cells were transfected with HA plasmid of H1N1 virus. Drug treatment with DMSO (0.1%), 5 μM 2-BP (positive control inhibitor), or 20 μM AM580 was carried out post-transfection, while cell lysates were harvested 24 h later. Total HA (input) and palmitoylated HA of different groups were analyzed using western blot. In all assays above, AM580 (20 μM) was added after virus absorption and maintained in the cell culture medium

aspects of viral life cycle and suggests that these aspects are vulnerable to antiviral intervention. SREBPs-mediated lipogenesis, affecting the building of membrane blocks, energy supply, and post-translational protein modification, is a complex process with cross talks and compensatory mechanisms that influence propagation of diverse groups of viruses (Fig. 7). Despite such complexities, DNA-binding activity of n-SREBPs is a potentially important target for therapeutic intervention against a broad spectrum of pathogenic viruses.

Tang et al.[22] previously reported that betulin inhibited maturation of SREBP by inducing interaction of SREBP cleavage activating protein (SCAP) and Insulin-induced gene protein

(Insig)[22]. Indeed, both betulin and a pre-SREBPs cleavage inhibitor PF 429242 exhibited some inhibitory effect on MERS-CoV replication at non-toxic concentrations (Supplementary Figure 4a). 25-Hydroxycholesterol (25HC), a cholesterol metabolite inhibiting infection by a broad range of pathogenic viruses, works indirectly through a 25HC sensor protein to elicit SCAP-Insig binding[29,30]. While AM580 acts differently from the above inhibitors, these studies supports that proteolytic processing of SREBPs may also be a feasible target for broad-spectrum antiviral therapy. Similar to these studies, the use of cancer cell lines which usually underwent complex genetic and metabolic rearrangement may affect the interpretation of results[31]. In addition, usage of

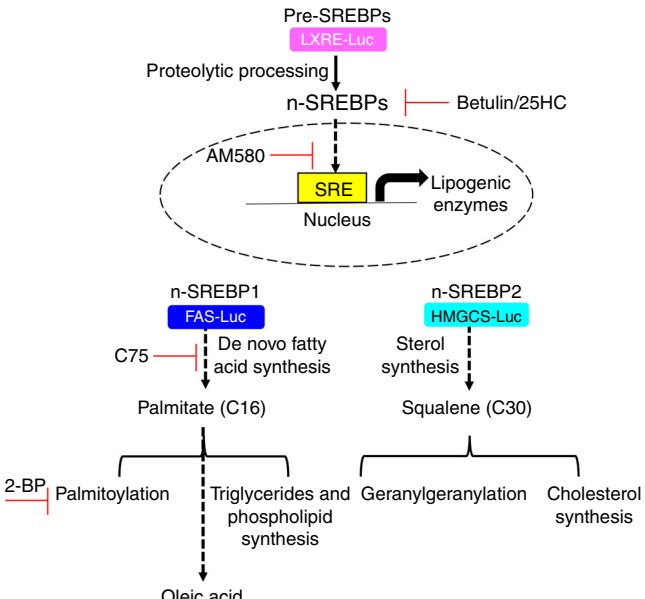

**Fig. 7** Schematic diagram showing the proteolytic processing of precursor SREBPs to generate n-SREBPs, downstream effector pathways (i.e. n-SREBP1-mediated fatty acid synthesis and n-SREBP2-mediated cholesterol synthesis), the positive control inhibitors and their pharmacological targets, and reporter gene constructs that were used in this study

odd carbon chain glycerophospholipid internal standards such as PC (12:0/13:0), PE (17:0/17:0)), PG (17:0/17:0), and addition of such standards directly to biological samples before lipid extraction would be more equitable and be used in the future lipidomic sample preparation[32].

The importance of SREBP regulatory pathway has been implicated in the life cycle of hantavirus which is dependent on lipid synthesis for virus entry and membrane fusion process[33,34]. This is also exemplified by the pharmacological inhibition of FAS by C75 or Cerulenin that impaired the replication of a broad-spectrum of viruses. Specifically, HCMV and influenza A virus infections may require fatty acid synthesis to modulate membrane composition for viral budding or protein modifications[35]. Direct inhibition of FAS reduces infectivity of respiratory syncytial virus and human rhinovirus serotype 16 by changing the membrane components of the virion progeny particles. DENV NS3 recruits FAS to sites of replication and stimulates fatty acid synthesis[36]. Distinct from those studies, our work underscores the indispensability of SREBP-mediated fatty acid synthesis to multiple post-virus-entry events, particularly for the emerging infectious viruses such as MERS-CoV and Influenza A virus. As examples, reduced DMV formation and viral protein palmitoylation are two important consequences of SREBP-controlled fatty acid synthesis inhibition (Fig. 6e, g). Studies on other viral replication processes related to SREBP-mediated downstream pathways are warranted. Furthermore, we demonstrated that the inhibition of diverse viruses, including enveloped (MERS-CoV, H1N1) and non-enveloped (EV-A71) RNA, and DNA (AdV5) viruses, could be partially rescued by exogenously supplied palmitate and to different extents (Fig. 6a, d). These results also support the requirement of de novo fatty acid synthesis for optimal viral replication. Moreover, transcriptional inactivation of lipogenic enzymes, in addition to FAS, may impair virus replication through diverse pathways in virus-specific manners (Fig. 7).

An unexpected result of the study is that antiviral activity of AM580 is neither dependent on the RAR-α pathway (Supplementary Figure 6a–c), nor on the activation of host innate immune response (Supplementary Figure 6d, e). Possible "off-target" effects of AM580 leading to its broad antiviral activity cannot be excluded as the compound was long known to be an RAR-α agonist. Moreover, a small-molecule compound can be a multi-target binder due to its intrinsic structural features, while the elicited biological responses through one of the ligand-target interactions found presently may not necessarily account for the observed antiviral activity. Although teratogenic effects of AM580, as a pro-differentiation agonist and not a cytotoxic, have been reported in mouse models[37], the anticipated therapeutic benefits outweigh its side effect when considering the short duration of 3-day treatment by AM580 in the animal experiments. In exploring AM580 analogs with comparable potency and improved bioavailability, tamibarotene[38], an orally active retinoid for treatment of acute promyelocytic leukemia was identified (Supplementary Table 1). Tamibarotene has a known safety profile and is marketed in Japan. With the clinical experience of tamibarotene, we believe AM580-related compounds could be modified as a safe and broad group of antiviral therapeutic compounds. Overall, by targeting the SREBP-associated lipid biosynthetic pathways, we demonstrate that the lipidomic reprogramming of host cells by various viruses exposes vulnerable targets for therapeutic intervention.

## Methods

**Cells and viruses.** Human embryonic kidney (HEK293T, ATCC, CRL-3216) cells, human lung carcinoma (A549, ATCC, CCL-185) cells, human hepatoma (Huh7, JCRB, 0403) cells, human rhabdomyosarcoma (RD, ATCC, CCL-136) cells, human epithelial type 2 (HEp-2, ATCC, CCL-23) cells, human lung adenocarcinoma (Calu-3, ATCC, HTB-55) cells, human leukemic (THP-1, ATCC, TIB-202) monocytes, Madin-Darby canine kidney (MDCK, ATCC, CCL-34) cells, African green monkey kidney (Vero, ATCC, CCL-81) cells, and Vero-E6 (ATCC, CRL-1586) cells were maintained in Dulbecco's Modified Eagle's medium (DMEM) culture medium supplemented with 10% heat-inactivated fetal bovine serum (FBS), 50 U/ml penicillin and 50 μg/ml streptomycin[39]. All cell lines were confirmed to be free of mycoplasma contamination by Plasmo Test (InvivoGen). Monocyte preparation and differentiation were performed according to an established protocol[40]. HSAEC (ATCC, PCS-301-010) were cultured with airway epithelial cell basal medium according to the manufacturer's protocol. Upon virus infection, infected cells were maintained in FBS-free medium with or without compounds. The MERS-CoV (HCoV-EMC/2012, a gift from Dr. Ron Fouchier) and SARS-CoV (GZ50) were propagated in Vero-E6 cells. The influenza A virus strain A/Hong Kong/415742/2009(H1N1)pdm09 and A/Anhui/1/2013(H7N9) were cultured in embryonated chicken eggs. Enterovirus A-71 (SZ/HK08-5) was cultured in RD cells. A clinical isolate of ZIKV (Puerto Rico strain PRVABC59, a gift from Dr. Brandy Russell and Dr. Barbara Johnson, Centers for Disease Control and Prevention, USA) was amplified in Vero cells. A clinical isolate of human adenovirus type 5 (AdV5) was propagated in A549 cells. All cultured viruses were titrated by PFU assays and/or 50% tissue culture infectious dose (TCID$_{50}$) assay and/or RT-qPCR assays[41]. All virus stocks were kept at −80 °C in aliquots. RT-qPCR assays were also applied for quantification of gene copies of interest using the Roche LightCycler Real-time PCR system. All primer sequences are provided in the Supplementary Table 2.

**Plasmids.** FAS-Luc containing FAS promoter was a gift from Dr. Bruce Spiegelman (Addgene plasmid#8890). pSynSRE-T-Luc (named as HMGCS-Luc for clarity) containing the HMGCS promoter (Addgene plasmid#60444), pcDNA3.1-2 × FLAG-SREBP-2 (Addgene plasmid#26807), pcDNA3.1-2 × FLAG-SREBP-1c (Addgene plasmid#26802) were gifts from Dr. Timothy Osborne. The luciferase constructs IFNβ-Luc and ISRE-Luc were provided by Dr. Dong-yan JIN (The University of Hong Kong). The LXRE reporter plasmid pGreenFire1-LXRE, named as LXRE-Luc in this study, was obtained from System Biosciences.

**Chemical and biological reagents.** AM580 was purchased from Cayman Chemical (Michigan, United States), Remdesivir was custom synthesized by WuXi AppTec (Tianjian, China), while other chemical inhibitors were obtained from Sigma-Aldrich (St. Louis, Missouri, United States) unless specified. Fluorescent neutral lipid dye 4,4-difluoro-1,3,5,7,8-pentamethyl-4-bora-3a,4a-diaza-s-indacene (BODIPY 493/503, Invitrogen) was used to stain LDs, while Filipin III (Cayman Chemical) was employed for visualization of intracellular cholesterol. The phosphine-activated fluorescent dye DyLight™ 488-Phosphine (invitrogen) was utilized for specific labeling and detection of azido-tagged molecules, i.e. azido-AM580. 4′,6-diamidino-2-phenylindole (DAPI, Sigma) and Phalloidin–Atto 647N

(Sigma) were used for nuclear and cell membrane staining, respectively. MERS-CoV NP was detected with the guinea pig anti-MERS-CoV NP serum (1:1000)[42]. Primary antibodies against human RAR-α (Abcam, ab41934, 1:2000), SREBP1 (Santa Cruz, sc-365513, 1:1000), SREBP2 (Proteintech, 14508-1-AP, 1:2000), anti-β actin (Abcam, ab8227, 1:2000), anti-influenza HA antibody (GeneTex, GTX127357, 1:1000) and Flag tag (Sigma, F3165, 1:1000) were used in relevant experiments. Uncropped western blottings are provided in Supplementary Figure 8. Alexa Fluor 488 goat anti-guinea pig IgG (H + L) antibody (Invitrogen, A-11073, 1:500) was utilized as a secondary antibody for immunofluorescence staining. Silencer Select human SREBP1 siRNA, Silencer Select human SREBP2 siRNA, and Silencer Select siRNA negative control were obtained from Life Technologies. Nuclear extraction was performed using commercial kit (Abcam) according to the manufacturer's protocol. SREBP1 transcription factor assay kit and SREBP2 transcription factor assay kit from Abcam were used for determination of SRE binding activity, respectively. Badrilla (Leeds, UK) kits were used for purification of palmitoylated proteins.

**Transcriptome analysis**. Calu-3 cells were mock infected or infected with MERS-CoV at 2 MOI and incubated in DMEM medium. At 24 hpi, total cellular RNAs of virus-infected or non-infected groups ($n = 3$) were collected. Gene expression following MERS-CoV infection was analyzed using RNA-Seq technology[43]. Sequencing libraries were constructed and sequenced by Beijing Genomics Institute (BGI), averagely generating 23,977,722 clean reads after filtering the low quality. Clean reads were mapped to reference sequences using HISAT/Bowtie2 (refs. [44,45]). The DEGs in MERS-CoV-infected samples were submitted to DAVID server for pathway enrichment and cluster analysis. Gene Ontology (GO) enrichment analysis of DEGs was implemented using the GOseq R package[46]. KOBAS software was used to examine the statistical enrichment of DEGS in The Kyoto Encyclopedia of Genes and Genomes (KEGG) pathways[47].

**Lipidome analysis**. Calu-3 cells were mock infected or infected with MERS-CoV at 2 MOI and incubated in DMEM medium. At 24 hpi, cells were collected for cellular lipid extraction. Inactivation of virus infectivity was confirmed before further processing using $TCID_{50}$ assay[48].

Liquid chromatography-mass spectrometry (LC-MS) grade water, methanol, and acetonitrile were purchased from J.T. Baker (Center Valley, PA, USA). HPLC-grade chloroform was purchased from Merck (Darmstadt, Germany). Analytical grade acetic acid and commercial standards used for biomarker identification were purchased from Sigma-Aldrich (MO, USA). Internal standards (IS) including Arachidonic acid-d8, 15(S)-HETE-d8, Leukotriene-B4-d4, and Lyso-Platelet-activating Factor C16-d4 (PAF C-16-d4) were purchased from Cayman (Cayman Chemical, USA).

The sample preparation was performed with minor modifications according to the Mplex method[49,50]. Medium was removed from infected cells at 24 hpi and the cells were immediately washed twice with ice-cooled quenching buffer consisted of 60% methanol and 0.85% ammonium bicarbonate in water. After removing the quenching buffer, 500 μl of ice-cold 150 mM ammonium bicarbonate solution was added to dissociate cells. The cells were then scraped and transferred into 15 ml tubes (Nunc, Thermo Fisher, USA) on ice. Fifty microliters of the cell suspension was removed for DNA extraction to perform cell count normalization. A genomic DNA mini-kit (QIAGEN, Germany) was used for all DNA extractions. Then 2 ml of lipid extraction solution (−20 °C chloroform/methanol; 2:1, v/v) containing IS was added to the tubes. A 100 μl aliquot of the mixed solution was removed for the confirmation of sterility by infectivity assay after lipid extraction. The remaining solution was centrifuged at 4500 rpm for 10 min at 4 °C and the bottom phase was transferred to a glass vial. All vials were then removed from BSC and BSL containment laboratory after appropriate disinfection and dried by a vacuum concentrator. The dried samples were reconstituted in 300 μl solvent mixture containing chloroform/methanol (v/v 2:1). After centrifugation at 14,000 rpm for 10 min at 4 °C, supernatants were transferred to LC vial for LC-MS analysis.

The cell lipid extract was analyzed using an Acquity UPLC system coupled to a Synapt G2-HDMS mass spectrometer system (Waters Corp., MA, USA). The chromatography was performed on a Waters ACQUITY BEH C18 column (1.7μm, 2.1 × 100 mm, I.D., 1.7 μm, Waters, Milford, MA, USA). The mobile phase consisted of (A) 0.1% acetic acid in water and (B) acetonitrile. The separation was performed at a flow rate of 0.4 ml/min under a gradient program as follows: 0.5% B (0–1.5 min), 0.5–8% B (1.5–2 min), 8–35% B (2–7 min), 35–70% B (7–13 min), 70–99.5% B (13–29 min), 99.5% B (29–36 min). In order to achieve rapid equivalence, the flow rate was changed to 0.5 ml/min after 36 min and the subsequent gradient program was applied as follows: 99.5% B (36–36.1 min), 99.5 to 0.5% B (36.1–38.1 min), 0.5% B (38.1–40 min). The column and auto-sampler temperature were maintained at 45 °C and 10 °C, respectively. The injection volume was 8 μl.

The mass spectral data were acquired in both positive and negative modes. The capillary voltage, sampling cone voltage, and source offset were maintained at 2.5 kV, 60 V, and 60 V, respectively. Nitrogen was used as desolvation gas at a flow rate of 800 l/h. The source and desolvation temperatures were maintained at 120 °C and 400 °C, respectively. Mass spectra were acquired over the $m/z$ range of 50–1200. The SYNAPT G2-Si HDMS system was calibrated using sodium format clusters and operated in sensitivity mode. Leucine enkephalin was used as a lock mass for

all experiments. The MS data acquired mode are $MS^E$ for lipids profile and MS/MS acquisition was operated in the same parameter as in MS acquisition for lipids identification. Collision energy was used with the range from 20 to 40 eV for fragmentation to allow putative identification and structural elucidation of the significant lipids.

Acquisition of the raw data was performed using MassLynx software version 4.1 (Waters Corp., MA, USA) and these raw data were firstly converted into the Analysis Base File (ABF) format. Then converted data were subsequently deconvolved into mass feature (MF) list using the MS-DIAL software (http://prime.psc.riken.jp/, version 2.46)[51,52]. Processed data were then exported as a text file for further statistical analysis. The total ion intensities of all MFs in the MS-DIAL report file for each sample were normalized by the corresponding DNA concentration to calibrate for cell count variations[53]. MetaboAnalyst 4.0 (http://www.metaboanalyst.ca) was used for univariate and multivariate statistical analysis for the normalized MFs subjected to the Pareto scaling method[54]. For univariate analysis, statistical significance of features was determined among the mock, MERS-CoV infected, and AM580-treatment groups using the Student's $t$-test, respectively. The $p$-value <0.05 were considered to be statistically significant features. For multivariate analysis, orthogonal partial least-squares discriminant analysis (OPLS-DA) was performed as a supervised method to find important variables with discriminative power and the model was evaluated with the relevant R2 and Q2.

The protocol of lipid identification was carried out according to published procedure[51,55]. The statistically significant features were given annotation names by MS-DIAL once the MS features could match internal library criterion. MFs with significant abundance were selected for MS/MS experiment and analysis. All putative lipids were identified by using exact molecular weights, nitrogen rule, MS2 fragment, and literature/database searches including METLIN database (http://metlin.scripps.edu/), Human Metabolome Database (http://www.hmdb.ca/), LipidMaps (http://www.lipidmaps.org/), and KEGG database (http://www.genome.jp/kegg). For final confirmation of lipids identity using authentic chemical standard, MS/MS fragmentation pattern of the chemical standard was compared with that of candidate lipids under the same LC-MS condition.

**Integrative omics study**. The joint lipidomic and transcriptomic analysis was done by MetaboAnalyst (version 4.0)[56], which enabled simultaneously mapping of upregulated genes and changed lipid after MERS-CoV infection.

**Lipid library screening**. The library with 189 compounds (Cayman Chemical, Michigan, USA), including prostaglandins, receptor agonists and antagonists, and ceramide derivatives, which is ideal for G protein-coupled receptor screening, was used for pharmacological screening. An MTT-based CPE inhibition assay was performed to identify compounds that could protect cells from virus infection[57]. To identify anti-MERS-CoV inhibitors, confluent Huh7 cells in 96-well culture plates ($4 \times 10^4$ cells/well) in triplicates were infected with MERS-CoV at 0.1 MOI. One hour after virus absorption, the inoculum was removed, followed by addition of drug-containing medium (10 μM). Twenty-four hours later, 10 μl of 5 mg/ml MTT solution (Sigma) were added to the wells. The monolayers were incubated for 4 h. Finally, 100 μl of 10% SDS with 0.01 M HCl was added and samples were further incubated at 37 °C in the presence of 5% $CO_2$ overnight. Color changes were read at $OD_{570}$ with a reference wavelength at $OD_{640}$. To screen anti-H1N1 inhibitors, MDCK cells were infected with H1N1 at 0.01 MOI, with cell viability scored at 48 hpi; other procedures were the same as for anti-MERS-CoV screening. Remdesivir (5 μM)[58] and Favipiravir (50 μg/ml)[59] were used as a positive control for MERS-CoV and H1N1, respectively. Next, a dose–response analysis using PRA[60] was performed to evaluate the in vitro antiviral efficacies of the primary hits, in which individual compound was serially diluted (10, 5, 2.5, 1.25, and 0.625 μM) and tested for either anti-MERS-CoV or anti-H1N1 activity.

**Selectivity index**. Selectivity index (SI) of the compound was calculated as the ratio of 50% cellular cytotoxicity concentration ($CC_{50}$) over 50% inhibitory concentration ($IC_{50}$). $CC_{50}$ value was determined with an MTT assay and by CellTiter-Glo assay (Promega) according to the manufacturer's protocol. $IC_{50}$ data were obtained with PRA or by viral load reduction assay as indicated[51]. Both $CC_{50}$ and $IC_{50}$ were calculated using GraphPad Prism 7.

**Flow cytometry**. For intracellular staining, cells were detached with 10 mM EDTA in PBS, fixed in 4% paraformaldehyde, and permeabilized with 0.1% Triton X-100 in PBS. Immunostaining for flow cytometry was performed following standard procedures[62]. Flow cytometry was performed using a BD FACSCanto II flow cytometer (BD Biosciences) and data were analyzed using FlowJo vX (Tree Star). The gating strategy is provided in Supplementary Figure 9.

**Human intestinal organoid culture and virus infection**. Intestinal organoids (intestinoids) were then cultured and differentiated for MERS-CoV infection by Ex vivo culture technology[18]. An inoculum of $10^5$ PFU of MERS-CoV was used to infect one droplet of intestinoids, with an estimated MOI of 0.1. After the inoculum was removed, the virus-inoculated intestinoids were rinsed with PBS and then re-embedded in Matrigel and cultured in a 48-well plate with culture medium

containing or lacking AM580 (20 μM). At indicated time points, intestinoids were harvested for the quantification of intracellular viral load, whereas the cell-free Matrigel and culture medium were combined for viral titration of extracellular virions using a standard plaque assay.

**Mouse experiments**. MERS-CoV and H7N9 virus were tested in hDDP4 Tg mice[18] and BALB/c mice[20], respectively. To examine the anti-MERS-CoV activity of AM580, a total of 56 mice (28mice/group) were evaluated. After anesthesia, mice were intranasally (i.n.) inoculated with 20 μl of virus suspension containing 50 PFU of MERS-CoV. Intraperitoneal (i.p.) therapeutic treatments were initiated 6 h post-virus-challenge. One group of mice was inoculated with 200 μl of AM580 i.p. for 3 days (12.5 mg/kg/day). The untreated control group of mice was administered 200 μl 0.1% DMSO in PBS i.p. Animal survival and clinical disease were monitored for 14 days or until death. Four mice in each group were euthanized randomly on day 2 and 4 post-challenge, respectively. Mouse lungs and brains were collected for virus titration and H&E histopathologic analyses[20]. To evaluate the anti-influenza virus potency of AM580 in vivo, BALB/c mice (18 mice/group) were inoculated i.n. with 100 PFU of H7N9 virus in 20 μl. Treatment was initiated 6 h post-virus challenge by i.n. administration. One group of mice was inoculated with 20 μl of i. n. AM580 (1 mg/kg/day). A second group was treated with 20 μl of i.n. zanamivir (2 mg/kg/day) as a positive control. A third group was given i.n. 0.1% DMSO in PBS as an untreated control. Two i.n. doses per day of AM580, zanamivir, or DMSO were administered for 3 days (total 6 doses/mouse). Animal survival and clinical disease were monitored for 14 days or until death. Lung tissues (4 mice/ group) were collected for viral load detection and H&E histopathologic analyses on days 3 and 6 post-virus-challenge, respectively.

**Molecular docking**. The crystal structures of SREBP1 (PDB code: 1AM9) and SREBP2 (PDB code: 1UKL) were retrieved from the Protein Data Bank database. SREBP1 dimer and SREBP2 dimer were separated with Pymol. Missing residues in SREBP2 were modeled using I-TASSER server[63]. Protein models were prepared with the Protein Preparation Wizard module in Maestro[64]. The 3D conformer of AM580 was downloaded from PubChem database[65]. Leadfinder v 1.81 was used to perform the docking simulation with extra precision method[66]. Alignment of the amino acid sequence was performed using Vector NTI (Thermo Fisher Scientific).

**Chemical synthesis**. Azido-AM580 was used for intracellular visualization of AM580, while AM580dp was designed and synthesized for pull-down of AM580 binding targets. To synthesize azido-AM580, 20 mg of AM580 was mixed with 3.14 ml of azido-PEG5-amine (10 mg/ml) dissolved in dimethylformamide (DMF). Next, 40 mg of 1-[bis(dimethylamino)methylene]-1H-1,2,3-triazolo[4,5-b]pyr-idinium 3-oxid hexafluorophosphate, 50 μl of N,N-diisopropylethylamine (DIPEA), and 2.46 ml of dichloromethane were added to a final volume 5.7 ml. Reactions were performed at room temperature with shaking. After overnight incubation, the reaction mixture was lyophilized to remove solvent. Azido-AM580 was purified by HPLC and m/z 640 was detected by mass spectrometry (MS). Final yield was quantified by nuclear magnetic resonance (NMR). To synthesize AM580dp, the purified azido-AM580 was linked with an amine reactive tri-functional crosslinker (2-[N2-[Nα-Benzoylbenzoicamido-N6-6-biotinamidoca-proyl]lysinylamido]ethyl-2′-(N sulfosuccinimidylcarboxy)ethyl disulfide sodium salt (Santa Cruz), which was a biotin-UV activating-N-hydroxysuccinimide (NHS)-ester compound. This tri-functional crosslinker contains a NHS-ester head group for linking to the amine tail of another crosslinker phosphine compound methyl 4-[2-[2-(2-aminoethoxy)ethoxy]ethylcarbamoyl]-2-diphenylphosphanyl-benzoate (Shinsei Chemical Company Ltd), a biotin head group for Streptavidin binding on the other end, and a UV activating benzophenone group for crosslinking with target binding proteins of AM580. Specifically, 1 mg of tri-functional linker compound was mixed with 30 μl of 74.8 mM crosslinker phosphine compound (dis-solved in DMSO-d6) to a final volume of 1 ml by DMF. The molar ratio of these compounds was 1:2. The reaction was performed at 40 °C with shaking at 1400 rpm. Excess azido-AM580 was added to this mixture to allow crosslinking with the phosphine group by Staudinger ligation reaction. The reaction product was then incubated with Streptavidin agarose resin (Pierce) to capture the AM580dp product.

**Electron microscopy**. Electron microscopy was utilized to observe DMVs induced by MERS-CoV infection. Vero cells were grown in six-well plates. Following infection with MERS-CoV at 3 MOI for 1 h, the cell culture medium was replaced with fresh medium containing 20 μM AM580 or 0.1% DMSO as a control. After 12 h, the cell culture medium was removed. The cells were washed with PBS, tryp-sinized, and fixed with glutaraldehyde for further processing and counterstaining[67]. The images were acquired in a Philips CM100 Transmission Electron Microscope located in Electron Microscope Unit of the University of Hong Kong.

**Statistics**. As indicated in the figure legends, each experiment was performed independently at least three times. For each experiment using samples collected from mice, a whole experimental set used for different treatments was harvested from a single animal. Statistical analyses were conducted using GraphPad Prism (version 7.0). Error bars indicated standard deviations. P-values were calculated

using either the Students' t-test, or one-way ANOVA, or Log-rank (Mantel–Cox) test as indicated. $*p < 0.05$, $**p < 0.01$, $***p < 0.001$.

**Study approval and biosafety measures**. The major usage of experimental models and subjects involved in this study was to evaluate the broad-spectrum antiviral effects of AM580 in multiple cell lines, mouse models, human primary cells, and organoid systems. Under the protocol approved by the Institutional Review Board of The University of Hong Kong/Hospital Authority Hong Kong West Cluster, normal small intestine tissue from a patient was obtained surgically, and human peripheral blood macrophages (MDMs) were isolated from healthy blood donors at Hong Kong Red Cross Blood Transfusion Service. Informed consent was obtained from all human participants and the experiments were performed in compliance with the approved standard operating procedures. Human dipeptidyl peptidase 4 (hDPP4) transgenic C57BL/6 mice (male and female) and BALB/c female mice were kept in biosafety level 3 housing and given access to standard pellet feed and water ad libitum. All experimental protocols were approved by the Animal Ethics Committee in the University of Hong Kong and were performed according to the standard operating procedures of the biosafety level 3 animal facilities. All experiments with live viruses were conducted using biosafety level 2 or 3 facilities[20]. The CULATR follows Hong Kong legislation and Association for Assessment and Accreditation of Laboratory Animal Care Inter-national recommended standards/guidelines (http://www.aaalac.org/about/ guide-lines. cfm).

**Reporting Summary**. Further information on experimental design is available in the Nature Research Reporting Summary linked to this article.

## Data availability
All relevant data are available from the authors upon request. Lipidomics data have been deposited to the EMBL-EBI MetaboLights database with the identifier MTBLS762. Transcriptomic data are available in GEO data repository under accession code GSE122876.

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

## Acknowledgements

This work was partly supported by the donations of Michael Seak-Kan Tong, the Shaw Foundation Hong Kong, Richard Yu and Carol Yu, Respiratory Viral Research Foundation Limited, Hui Ming, Hui Hoy and Chow Sin Lan Charity Fund Limited, Chan Yin Chuen Memorial Charitable Foundation, and the Hong Kong Hainan Commercial Association South China Microbiology Research Fund; and funding from the Theme-based Research Scheme (T11-707/15-R) of the Research Grants Council, Hong Kong Special Administrative Region; the Health and Medical Research Fund (HKM-15-M03, HKM-15-M04, HKM-15-M05, HKM-15-M11, and 15140762) of the Food and Health Bureau, Hong Kong Special Administrative Region Government; the Consultancy Service for Enhancing Laboratory Surveillance of Emerging Infectious Diseases for the HKSAR of the Department of Health, Hong Kong Special Administrative Region; the High Level Hospital-Summit Program in Guangdong, The University of Hong Kong-Shenzhen Hospital; and the National Institutes of Health (USA, PO1 AI060699 (SP)). The sponsors

had no role in the design and conduct of the study, in the collection, analysis and interpretation of data, or in the preparation, review or approval of the manuscript.

## Author contributions

S.Y., H.C., J.F.-W.C. and K.-Y.Y. designed the study. Z.-W.Y., J.H., D.C., V.K.-M.P., K.-M.T., M.C.C. and C.Y. performed experiments and collected analyzed data. L.W. analyzed the RNA-Seq results and did molecular docking. B.Y., P.-M.L., D.Y. and K.-H.S. performed the lipidome analysis and chemical synthesis. J.F.-W.C. and C.C.-S.C. provided the animal experiment data. J.Z., C.L., I.H.-Y.C. and X.Z. provided the organoid data. J.Y.-N.L. gave conceptual advice on structure-activity analysis. H.C., K.-H.K., K.K.-W.T., R.Y.-T.K., D.-Y.J. and S.P. interpreted the results and gave advice. S.Y., J.F.-W.C. and K.-Y.Y. supervised the study. S.Y., J.F.-W.C., S.P. and K.-Y.Y. wrote the manuscript.

## Additional information

**Competing interests:** The authors declare no competing interests.

