## [Peer Review File · Nature Communications]

Reviewers' Comments:

Reviewer #1:

Remarks to the Author:

MERS/SARD are deadly coronaviruses and currently there are no antivirals or vaccines available. Thus the study here, identification of a small lipid active molecule AM580 that has broad inhibitory activity against a variety of viruses whose replication is lipid dependent is an important and timely one. The effective dose of this small molecule appears to be well below the doses that are cytotoxic. In vivo data in mice are also quite promising. Collectively these are all exciting findings worthy of publication and further pursuit.

One issue is whether the mechanism of action is really through SREBP: siRNA knockdown of SREBP1 or SREBP2 only show a log decrease in viral copy number whereas the infected mice treated with the molecule appear to be cured. Have the authors tried a double knock down of SREBP1 and SREBP2? or CRISPR ?

Reviewer #2:

Remarks to the Author:

Summary: In this manuscript, the authors use transcriptomics and lipidomics of MERS-CoV infected cells to identify alterations in cellular lipid metabolism. They then screen a bioactive lipid library and identify a lipid (AM850) that has broad spectrum antiviral activity in vitro and in vivo. Characterization of this compound suggests that it targets nSREBP, a common node of viral infection. Docking and binding studies lend support for the AM850-SREBP interaction. This impacts lipid biosynthesis, and they show a role for inhibition of fatty acid synthesis (FAS) in particular.

I am generally supportive of this manuscript. Although the SREBP/FAS pathways have been implicated as both virally manipulated and a broad-spectrum drug target in previous studies, the identification of a bioactive lipid inhibitor with efficacy in vitro and in vivo plus a good selectivity index is quite interesting. Some experiments lack controls and many are over-interpreted. This needs to be addressed in revision.

Specific points:

1. As mentioned above, many studies have identified SREBP/FAS as virally manipulated and that inhibiting them inhibits viral replication (including some of the viruses in this study such as IAV and flaviviruses). These should be referenced and discussed.
2. A caveat of Fig. 1 transcriptomic/lipidomic analysis is that it is done in a cancer cell line that already has deregulated metabolism. This caveat needs to be mentioned.
3. The abstract is misleading. It suggests that you used an unbiased Click approach to identify SREBP, when in fact you used a heavily biased candidate approach to investigate SREBP. The abstract needs to be edited for accuracy or better, perform the Click-MS proteomics approach to see what AM850 binds to in an unbiased fashion.
4. Fig. 3b,f differences are not overly impressive. Perform statistical analysis and indicate if there is a significant difference in the times prior to sacrifice.
5. Fig. 4A-D and Fig. 6E have issues with controls and interpretation. It is not clear whether the phenotypes result directly from inhibition of a cellular process or indirectly by inhibiting viral

replication. For instance, does AM850 affect lipid droplet (LD) accumulation directly or is there just less viral enhancement of LDs because there is less viral replication? Are there no DMVs because of a direct role in DMV formation or because replication is inhibited? A way around this issue is to create cell lines expressing viral proteins/polyproteins responsible for alterations such that their expression is unaffected by the drug. This approach has been used extensively in related virus fields. The authors could also investigate AM850 in uninfected cells or cells inducing lipid metabolism via other means. Currently the data is way over-interpreted.

Other issues include a lack of RNAi specificity controls (multiple distinct siRNAs or preferably, expression of siRNA resistant SREBPs). Fig. 4D is also over-interpreted. Starvation does more than induce SREBPs, such as inducing autophagy. This experiment needs to be removed and replaced by SREBP over-expression.

6. Fig. S4 lacks vehicle only (usually DMSO) controls and cell viability controls.

7. Fig. 6 has two issues. The smaller one is "paltimate" in the graph legend in 6A-D. The second is that just because there is less palmitoylated HA does not mean this is why infection is inhibited. Lipids downstream of palmitate may be important. In your complementation experiments, you should also include a downstream lipid such as oleic acid. If your model is correct, the oleic acid should complement the (+)RNA viruses, but not IAV.

Reviewer #3:

Remarks to the Author:

The paper presents data that viral infection alters the lipidomic profile of the host cells and that the synthetic Retinoic Acid Receptor α (RAR α) agonist AM580 interrupts the virus life cycle. Also, it was demonstrated that AM580 selectively binds to Sterol Regulatory Element Binding Proteins (SREBPs). While the authors showed a heatmap of the lipids that are purportedly changed with viral infection, the details of the method and the identification of the lipids are extremely sketchy to evaluate the analytical methodology. For example, how are the lipids identified? There was neither a description of the identification or a reference to a method. The lipids were presumably extracted by standards Bligh-Dyer method and the extract was analyzed by LC-MS. In the procedure, the authors describing the reconstitution of the dried lipid extract in chloroform-methanol (2:1) but injected directly on LC-MS with initial mobile phase conditions that essentially consist of water (containing 0.5% acetonitrile and 0.1% acetic acid) at a flow rate of 0.4 ml/min. Despite the presence of methanol in the sample, the sample will be immiscible with the mobile phase at that flow rate. How were they able to perform the chromatography? There was no mention of any internal standards used for LC-MS. How are the lipids quantified to ascertain the changes upon infection? The lipids identified in Figure 1b do not conform to any standard nomenclature nor was there any key in Supplemental data to identify the lipids. More details are needed to evaluate the method and the results presented.

If AM580 was able to inhibit virus replication, does the treatment of virus infected cells with AM580 restore lipidomic profile? If not, the link between AM580's ability to inhibit virus replication and lipidomic profile changes are circumstantial at best. In the data presented on the screening of commercial lipid library to inhibit virus replication and enhance cell viability, there are at least two other compounds that are either equal or better than AM580. What are these lipids and do they also bind to SREBPs to inhibit virus replication? Despite the liberal use of the term 'bioactive lipid' by the authors, AM580 is not an endogenous, naturally occurring bioactive lipid. It is a synthetic agonist of RAR α . Presumably, there could be other compounds in the 'lipid library' that are synthetic and calling them 'bioactive lipids' is misleading or a lack of understanding of what a bioactive lipid is.

Reviewer #4:

Remarks to the Author:

The manuscript by Yuan et al. describes the identification of sterol regulatory element binding protein (SREBP) as a key regulator for lipid biosynthesis and a potential cellular target for antiviral therapeutics. The authors used transcriptomics and lipidomic profiling as well as multiple molecular and biochemical methods to show that proteolytic processes and lipid biosynthesis pathways, including palmitoylation and double-membrane vesicles formation, are affected by SREBP. The latter are known to be essential for virus replication and particle formation. The authors found that a previously established compound (AM580) blocked SREBP binding to sterol regulatory elements (SREs). AM580 treatment post infection inhibited virus replication in vitro (MERS-CoV, SARS-CoV, H1N1, EV-A71, ZIKV and AvV5) and in vivo (mouse model for MERS-CoV and Influenza virus). The authors therefore claim to have identified a broad-spectrum antiviral drug.

The manuscript is overall concise and well written. The data generally support the in vitro and in vivo data. The identification of a key regulator of lipid biosynthesis is of interest, in particular, as viruses exploit this pathway to sequester by-products of virus replication (dsRNA) from immune recognition and to produce virus progeny.

However, the claim to have identified a broad-acting antiviral therapeutic is in my opinion exaggerated. Neither the applied organoid intestinal model nor the relatively artificial transgenic mouse model justifies the title of the paper, i.e. "Targeting the SREBP-dependent lipidomic reprogramming by virus infection for broad-spectrum therapeutic intervention". Importantly, off-target and other downstream effects resulting from AM580 treatment should be investigated. In particular, as the AM580-induced lack of DMV formation may affect innate immune regulation. Supplementary Figure 6 is not sufficient to exclude that AM580 affects immune regulation in context of MERS-CoV replication.

Major

1. In all Figures the authors should provide PFU/ml and virus genome equivalents/ml either in the main Figures or at least as supplemental information. As AM580 blocks DMV formation one would expect an increase of defective interfering particles.
2. The compound AM580 inhibits DMV formation which likely results in increased exposure of MERS-CoV-derived dsRNA to cellular helicases with subsequent activation of the interferon pathway. Please include experiments using virus infection and AM580 treatment followed by the assessment of immune activation by, for example, real time PCR or IFN ELISA.
3. Figure 2a/ Figure 2a legend & Page 7 line 3: How did the authors guarantee that the inhibitor or its downstream effects were not disturbing the PFU formation in the plaque assays? Increased IFN secretion may, for example, inhibit plaque formation as MERS-CoV is highly IFN sensitive. Please always indicate in the Figure legends when AM580 was applied to the cells. Which cells were used in Figure 2a exactly?
4. Figure 2f-g: Intestinal organoids are an interesting model but MERS-CoV is obviously a respiratory virus. Human airway epithelial cells would be the appropriate model in this case.

Minor

1. Figure 1: Please resize, the figure is hard to read.

2. Figure 1 and suppl. Figure 1: Please mention the MOI and conditions in the Fig. legend
3. Figure 1d and Suppl Figure 1b: Please include a reference for the use of Remdesivir as positive control for MERS-CoV.
4. Figure 1c: Why does the steroid pathway not show up here, while it showed up in the KEGG enrichment?
5. Suppl Fig. 1a: Only a few genes are enriched here (10 or less), whilst GO enrichment showed 94 enriched genes. Is 10/94 reflective for the proportion of steroid biosynthesis/total lipid biosynthesis?
6. Figure 2b: The time-points of the N protein detection are not explained. The kinetic in Fig 2a shows the time-points beyond 12 hours until 48 hours, the WB shows 0-9 hours post infection.
7. Figure 2c: The detection of N protein by IF is not convincing in the DMSO control panel. Please improve the Figure quality.
8. Figure 3c: Please also provide information regarding the PFU/ml as mentioned above.
9. Figure 6e: moi or time-point not provided in the Figure legend

Reviewer #1 (Remarks to the Author):

1. MERS/SARD are deadly coronaviruses and currently there are no antivirals or vaccines available. Thus the study here, Identification of a small lipid active molecule AM580 that has broad inhibitory activity against a variety of viruses whose replication is lipid dependent is an important and timely one. The effective dose of this small molecule appears to be well below the doses that are cytotoxic. In vivo data in mice are also quite promising. Collectively these are all exciting findings worthy of publication and further pursuit.

One issue is whether the mechanism of action is really through SREBP: siRNA knockdown of SREBP1 or SREBP2 only show a log decrease in viral copy number whereas the infected mice treated with the molecule appear to be cured. Have the authors tried a double knock down of SREBP1 and SREBP2? or CRISPR?

Response: Thanks to the insightful comment of the reviewer. We agree with the reviewer that knockdown of SREBP1 or 2 did not exhibit similar level of antiviral potency as that of AM580, which might be due to mutual compensation between SREBP1 and SREBP2 when either one protein was impaired (Shimano et al., 1997). When the SRE-ds DNA-binding activity of both n-SREBP1 and n-SREBP2 were blocked by AM580, there should be an additional/synergistic antiviral effect. This is supported by our observation that double knockdown of SREBP1 and SREBP2 in the additional experiment resulted in about 2 logs decrease of infectious virus particle when compared with the 1 log reduction of individual knockdown. Together with the comments from reviewer #2 regarding the RNAi specificity, we have now included experiments using distinct siRNA targeting the same SREBP1 (i.e. 1_1 and 1_2) and SREBP2 (i.e. 2_1 and 2_2). These results are now included in Fig.4c and lines 3-8 of page 12.

We have tried CRISPR knockout of either SREBP1 or SREBP2 before, which appeared to be detrimental to the host cells. This might be due to the essential cellular functions of the SREBP pathway which is in line with others' reporting that SREBP-1 deficiency is lethal in utero in 50–85% of knockout mice and that SREBP-2 deficiency is lethal during embryogenesis (Horton, Goldstein, & Brown, 2002).

Reviewer #2 (Remarks to the Author):

1. Summary: In this manuscript, the authors use transcriptomics and lipidomics of MERS-CoV infected cells to identify alterations in cellular lipid metabolism. They then screen a bioactive lipid library and identify a lipid (AM850) that has broad spectrum antiviral activity in vitro and in vivo. Characterization of this compound suggests that it targets nSREBP, a common node of viral infection. Docking and binding studies lend support for the AM850-SREBP interaction. This impacts lipid biosynthesis, and they show a role for inhibition of fatty acid synthesis (FAS) in particular.

I am generally supportive of this manuscript. Although the SREBP/FAS pathways have been implicated as both virally manipulated and a broad-spectrum drug target in previous studies, the identification of a bioactive lipid inhibitor with efficacy in vitro and in vivo plus a good selectivity index is quite interesting. Some experiments lack controls and many are over-interpreted. This needs to be addressed in revision.

Specific points:

As mentioned above, many studies have identified SREBP/FAS as virally manipulated and that inhibiting them inhibits viral replication (including some of the viruses in this study such as IAV and flaviviruses). These should be referenced and discussed.

Response: Thanks to the helpful comments of the reviewer. We agree with the reviewer and have now included the following paragraph in lines 9-15 page 18 of the discussion section:

The importance of SREBP regulatory pathway has been implicated in the life cycle of hantavirus which is dependent on lipid synthesis for virus entry and membrane fusion process (Kleinfelter et al., 2015; Petersen et al., 2014). This is also exemplified by the pharmacological inhibition of FAS by C75 or Cerulenin that impaired the replication of a broad-spectrum of viruses. Specifically, human cytomegalovirus and influenza A virus infections may require fatty acid synthesis to modulate membrane composition for viral budding or protein modifications (Munger et al., 2008). Direct inhibition of FAS reduces infectivity of respiratory syncytial virus and human rhinovirus serotype 16 by changing the membrane components of the virion progeny particles. DENV NS3 recruits FAS to sites of replication and stimulates fatty acid synthesis (Heaton et al., 2010).

2. A caveat of Fig. 1 transcriptomic/lipidomic analysis is that it is done in a cancer cell line that already has deregulated metabolism. This caveat needs to be mentioned.

Response: We agree with the reviewer and have included this comment in the discussion (lines 4-6 of page 18).

Similar to these studies, the use of cancer cell lines which usually underwent complex genetic and metabolic rearrangement may affect the interpretation of results (Kaur & Dufour, 2012).

3. The abstract is misleading. It suggests that you used an unbiased Click approach to identify SREBP, when in fact you used a heavily biased candidate approach to investigate SREBP. The abstract needs to be edited for accuracy or better, perform the Click-MS proteomics approach to see what AM850 binds to in

an unbiased fashion.

Response: We agree with the reviewer and have made clarification accordingly. Please refer to page 3 lines 7-9.

Using click chemistry, the overexpressed sterol regulatory element binding protein (SREBP) was shown to interact with AM580 which accounts for its broad-spectrum antiviral activity.

4. Fig. 3b,f differences are not overly impressive. Perform statistical analysis and indicate if there is a significant difference in the times prior to sacrifice.

Response: To address the reviewer's comment, additional mice experiment has been done and the method for statistical analysis has been modified. The new Fig.3b showed that mice in AM580-treated group exhibited significantly higher body weight ($p < 0.05$) than that of DMSO-treated group on days 4 and 5 post-infection of the MERS-CoV infected mice. In terms of influenza H7N9 mouse model (new Fig.3f), AM580-treated mice showed less body weight loss than that of DMSO-treated group from day 3 to day 7 post-infection.

The current survival rate and body weight results for MERS-CoV infection model ($n=20$) is pooled by previous round ($n=10$) and this additional round during manuscript revision ($n=10$). Statistically, for both Fig.3b and 3f, we transformed the body weight change of individual mouse as %, followed by mean% and SD% calculation of each day post-infection; while in the previous version of manuscript, we calculated the mean and SD value of all the mice on each day first, followed by transformation to %. We believe that the new approach of data analysis is more reasonable.

We have now revised the description in result (page 10 lines 3-4 and lines 16-17) and updated the method (page 27) accordingly.

5. Fig. 4A-D and Fig. 6E have issues with controls and interpretation. It is not clear whether the phenotypes result directly from inhibition of a cellular process or indirectly by inhibiting viral replication. For instance, does AM850 affect lipid droplet (LD) accumulation directly or is there just less viral enhancement of LDs because there is less viral replication? Are there no DMVs because of a direct role in DMV formation or because replication is inhibited? A way around this issue is to create cell lines expressing viral proteins/polyproteins responsible for alterations such that their expression is unaffected by the drug. This approach has been used extensively in related virus fields. The authors could also investigate AM850 in uninfected cells or cells inducing lipid metabolism via other means. Currently the data is way over-interpreted.

Response: We agree with the reviewer's comment. One recent publication clearly demonstrated that co-expression of the nsp3 and nsp4 of MERS-CoV were sufficient to induce the DMV formation (Oudshoorn et al., 2017). Therefore, we used AM580 to treat the Huh7 cells co-transfected with nsp3 (~209kDa) and nsp4 (~57kDa) for 24 hours. No significant changes were detected in the of expression levels of both viral proteins. The finding indicates that the reduced DMV formation is caused by the inhibition of lipid accumulation directly and not by the inhibition of viral replication indirectly.

We have added this set of results in page16 lines 2-7 and new Fig.6f of the manuscript.

6. Other issues include a lack of RNAi specificity controls (multiple distinct siRNAs or preferably, expression of siRNA resistant SREBPs). Fig. 4D is also over-interpreted. Starvation does more that induce SREBPs, such as inducing autophagy. This experiment needs to be removed and replaced by SREBP over-expression.

Response: To address the issue of RNAi specificity, we have purchased distinct siRNAs targeting the same SREBP1 (i.e. 1_1 and 1_2) or SREBP2 (2_1 and 2_2). Together with the comment from reviewer 1, we also demonstrated that double-knockdown of SREBP1 and SREBP2 caused less virus replication when compared with the knockdown alone. Please refer to legends in page 39 lines 35-39 and new Fig. 4c for detail.

In terms of the overexpression of n-SREBP1 and 2, we have performed the assay accordingly and showed in new Fig.4d that overexpression of n-SREBP1 or 2 diminished the antiviral potency of AM580. Please refer to its result description (page 12 lines 11-12) and legends (page 39 lines 41-44) for detail.

7. Fig. S4 lacks vehicle only (usually DMSO) controls and cell viability controls.

Response: We agree with the reviewer and have added the cell viability curve and DMSO control accordingly (new Fig. S4 copied below), while the 0 μ M of each line is the DMSO vehicle control that the reviewer referred to. Please refer to the legend of new Fig.S4 for detail (supplementary information page 4).

8. Fig. 6 has two issues. The smaller one is “paltimate” in the graph legend in 6A-D. The second is that just because there is less palmitoylated HA does not mean this is why infection is inhibited. Lipids

downstream of palmitate may be important. In your complementation experiments, you should also include a downstream lipid such as oleic acid. If your model is correct, the oleic acid should complement the (+) RNA viruses, but not IAV.

Response: We apologize for the typo and have revised it as ‘palmitate’ accordingly. We have also included 100μM oleic acid in the complementation assays and demonstrated the inability of this fatty acid metabolite to rescue the influenza A H1N1 virus replication, which suggests it is the reduced influenza hemagglutinin palmitoylation that mainly account for the antiviral activity of AM580. Please refer to the new Fig. 6a-d and relevant results (page 16, lines 20-23) and legends (page 40, lines 33-38) for details.

Meanwhile, we have updated the role of oleic acid as a downstream molecule of palmitate (C16) in new Fig.7 schematic diagram to facilitate the interpretation of the results by readers.

Reviewer #3 (Remarks to the Author):

Comment 1

The paper presents data that viral infection alters the lipidomic profile of the host cells and that the synthetic Retinoic Acid Receptor α (RAR α) agonist AM580 interrupts the virus life cycle. Also, it was demonstrated that AM580 selectively binds to Sterol Regulatory Element Binding Proteins (SREBPs). While the authors showed a heatmap of the lipids that are purportedly changed with viral infection, the details of the method and the identification of the lipids are extremely sketchy to evaluate the analytical methodology. For example, how are the lipids identified? There was neither a description of the identification or a reference to a method. The lipids were presumably extracted by standards Bligh-Dyer method and the extract was analyzed by LC-MS. In the procedure, the authors describing the reconstitution of the dried lipid extract in chloroform-methanol (2:1) but injected directly on LC-MS with initial mobile phase conditions that essentially consist of water (containing 0.5% acetonitrile and 0.1% acetic acid) at a flow rate of 0.4 ml/min. Despite the presence of methanol in the sample, the sample will be immiscible with the mobile phase at that flow rate. How were they able to perform the chromatography? There was no mention of any internal standards used for LC-MS. How are the lipids quantified to ascertain the changes upon infection? The lipids identified in Figure 1b do not conform to any standard nomenclature nor was there any key in Supplemental data to identify the lipids. More details are needed to evaluate the method and the results presented.

Response: Thanks for the reviewer's very useful comments. To fully address the reviewer's concern, we have now divided the comments into several sub-questions as below:

Q1: How are the lipids identified? There was neither a description of the identification or a reference to a method. The lipids were presumably extracted by standards Bligh-Dyer method and the extract was analyzed by LC-MS.

Response: We apologize for the sketchy methodology provided in the previous version of manuscript while trying to reduce the number of words. We have now updated the manuscript including all the details of the lipidomics methodology, data analysis and Heatmap analysis as followed:

(1) Chemicals and reagents

Liquid chromatography-mass spectrometry (LC-MS) grade water, methanol and acetonitrile were purchased from J.T. Baker (Center Valley, PA, USA). HPLC-grade chloroform was purchased from Merck (Darmstadt, Germany). Analytical grade acetic acid and commercial standards used for biomarker identification were purchased from Sigma-Aldrich (MO, USA). Internal standards (IS) including Arachidonic acid-d8, 15(S)-HETE-d8, Leukotriene-B4-d4 and Lyso-Platelet-activating Factor C16-d4 (PAF C-16-d4) were purchased from Cayman (Cayman Chemical, USA).

(2) Sample preparation

The sample preparation was performed with minor modifications according to the Mplex method (Burnum-Johnson et al., 2017; Nakayasu et al., 2016). Medium was removed from infected cells at 24hpi and the cells were immediately washed twice with ice-cooled quenching buffer consisted of 60% methanol and 0.85% ammonium bicarbonate in water. After removing the quenching buffer, 500 μ l of ice-cold 150mM ammonium bicarbonate solution was added to dissociate cells. The cells were then scraped and transferred into 15ml tubes (Nunc, Thermo Fisher, USA) on ice. 50 μ l of the cell suspension was

removed for DNA extraction to perform cell count normalization. A genomic DNA mini-kit (QIAGEN, Germany) was used for all DNA extractions. Then 2ml of lipid extraction solution (-20°C chloroform/methanol; 2:1, v/v) containing IS was added to the tubes. A 100µl aliquot of the mixed solution was removed for the confirmation of sterility by infectivity assay after the lipid extraction. The remaining solution was centrifuged at 4500rpm for 10min at 4°C and the bottom phase was transferred to a glass vial. All vials were then removed from BSC and BSL containment laboratory after appropriate disinfection and dried by vacuum concentrator. The dried samples were reconstituted in 300µl solvent mixture containing chloroform/methanol (v/v 2:1). After centrifugation at 14000rpm for 10min at 4°C, supernatants were transferred to LC vial for LC-MS analysis.

(3) UPLC-ESI-Q-TOF-MS analysis

The cell lipid extract was analyzed using an Acquity UPLC system coupled to a Synapt G2-HDMS mass spectrometer system (Waters Corp., MA, USA). The chromatography was performed on a Waters ACQUITY BEH C18 column (1.7µm, 2.1×100mm, I.D., 1.7mm, Waters, Milford, MA, USA). The mobile phase consisted of (A) 0.1% acetic acid in water and (B) acetonitrile. The separation was performed at a flow rate of 0.4 ml/min under a gradient program as follows: 0.5% B (0 to 1.5min), 0.5 to 8% B (1.5 to 2min), 8 to 35% B (2 to 7min), 35 to 70% B (7 to 13min), 70 to 99.5% B (13 to 29min), 99.5% B (29 to 36min). In order to achieve rapid equivalence, the flow rate was changed to 0.5mL/min after 36mins and the subsequent gradient program was applied as followed: 99.5% B (36 to 36.1min), 99.5 to 0.5% B (36.1 to 38.1min), 0.5% B (38.1 to 40min). The column and auto-sampler temperature were maintained at 45°C and 10 °C, respectively. The injection volume was 8µl.

The mass spectral data were acquired in both positive and negative modes. The capillary voltage, sampling cone voltage and source offset were maintained at 2.5kV, 60V, and 60V, respectively. Nitrogen was used as desolvation gas at a flow rate of 800 L/h. The source and desolvation temperatures were maintained at 120 °C and 400 °C, respectively. Mass spectra were acquired over the m/z range of 50 to 1200. The SYNAPT G2-Si HDMS system was calibrated using sodium format clusters and operated in sensitivity mode. Leucine enkephalin was used as a lock mass for all experiments. The MS data acquired mode is MS^E for lipids profile and MS/MS acquisition was operated in the same parameter as in MS acquisition for lipids identification. Collision energy was used with the range from 20 to 40 eV for fragmentation to allow putative identification and structural elucidation of the significant lipids.

(4) Data processing and statistical data analysis

Acquisition of the raw data was performed using MassLynx software version 4.1 (Waters Corp., MA, USA) and these raw data were firstly converted into the Analysis Base File (ABF) format. Then converted data were subsequently deconvolved into mass feature (MF) list using the MS-DIAL software (<http://prime.psc.riken.jp/>, version2.46) (Kind et al., 2013; Tsugawa et al., 2015). Processed data were then exported as a text file for further statistical analysis. The total ion intensities of all MFs in the MS-DIAL report file for each sample were normalized by the corresponding DNA concentration to calibrate for cell count variations (Silva et al., 2013). MetaboAnalyst 4.0 (<http://www.metaboanalyst.ca>) was used for univariate and multivariate statistical analysis for the normalized MFs subjected to Pareto scaling method (Chong et al., 2018). For univariate analysis, statistical significance of features was determined among the mock-infected, MERS-CoV infected and AM580-treated MERS-CoV infected groups using the Student's t-test, respectively. The p value < 0.05 were considered to be statistically significant features. For multivariate analysis, orthogonal partial least squares discriminant analysis (OPLS-DA) was performed as a supervised method to find important variables with discriminative power and the model was evaluated with the relevant R² and Q².

(5) Lipids Identification

The protocol of lipid identification was carried out according to published procedure (Tsugawa et al., 2015; Zhu et al., 2013). The statistically significant features were given annotation names by MS-DIAL

once the MS features could match internal library criterion. MFs with significant abundance were selected for MS/MS experiment and analysis. All putative lipids were identified by using exact molecular weights, nitrogen rule, MS2 fragment and literature/database searches including METLIN database (<http://metlin.scripps.edu/>), Human Metabolome Database (<http://www.hmdb.ca/>), LipidMaps (<http://www.lipidmaps.org/>) and KEGG database (<http://www.genome.jp/kegg>). For final confirmation of lipids identity using authentic chemical standard, MS/MS fragmentation pattern of the chemical standard was compared with that of candidate lipids under the same LC-MS condition.

In the current comparative lipidomics study, we focus on identifying significantly changed lipids among mock-infected cells, MERS-CoV-infected cells and MERS-CoV-infected cells with AM580 treatment. Supplementary Table S2 summarized the detail information of annotated significantly changed lipids, which included the fold changes and Student's t-test p values of MERS-CoV infected cells vs non-infected cells (MERS-CoV/MOCK) and AM580 treated MERS-CoV infected vs DMSO-treated MERS infected cells (AM580/MERS-CoV). It should be noted that, in addition to the annotated lipids reported in Table S2, there are a number of unknown features that exhibited significant cellular level changes but we could not identify. We reported these as preliminary identified lipids, which at least match the accurate mass on the MS-DIAL internal databases. For some lipids, we performed MS/MS fragmentation but no available fragment pattern was reported in the website databases or no diagnostic fragment patterns were available. These lipids could not be fully identified thus we labeled them as MS, which means they were only matched by the accurate mass with MS-DIAL internal database and website database, such as the FA(22:1), FAHFA(20:0), MGDG(30:1). Moreover, for those sphingolipids and glycerophospholipids where their number of acyl chain carbons and double bonds could not be determined, we labeled them with their total number of carbons and double bonds, such as the SM(34:0) and PE(38:5).

We have updated those experimental details in the methodology (pages 22-25) and please also refer to Supplementary Table 2 for the details of the results (Supplementary information, page 11).

Q2: In the procedure, the authors describing the reconstitution of the dried lipid extract in chloroform-methanol (2:1) but injected directly on LC-MS with initial mobile phase conditions that essentially consist of water (containing 0.5% acetonitrile and 0.1% acetic acid) at a flow rate of 0.4 ml/min. Despite the presence of methanol in the sample, the sample will be immiscible with the mobile phase at that flow rate. How were they able to perform the chromatography?

Response: Thank you for the reviewer's expert comment. We note that our LC gradient and mobile phase compositions are not optimal to cover all lipids classes even though we have used the mixed solution consist of chloroform-methanol (2:1) for lipid extraction. Our current LC protocol is designed for the analysis of polar lipids and was based on the LipidMaps (<http://www.lipidmaps.org/>) methods with modifications (Ivanova, Milne, Byrne, Xiang, & Brown, 2007). We have been able to detect large number of lipid mass features including fatty acids, glycerophospholipids and sphingomyelins from our infected and non-infected cell samples, and even in patient samples in previous studies, using the same methodology (Lau et al., 2016; K. K. To et al., 2016; K. K. W. To et al., 2015). More importantly, the intracellular membranes were mainly consisted of glycerophospholipids, which play important roles in signaling and viral replication cycle. So in the current study, we have focused on these glycerophospholipids changes based on the previously established lipidomics approach. Since acetonitrile can dissolve a wide range of ionic and nonpolar compounds, with the progressive increase of the acetonitrile content in the mobile phase, the present LC-gradient could provide good separations of the targeted lipid class. As an example, the 68 significantly changed lipid features displayed in

Supplementary Table 2 has shown that different types of fatty acids, glycerophospholipids and sphingomyelins were successfully separated by the present LC-gradient and were eluted at different retention times. Figure below showed the XIC curves of 12 representative lipid features in the present study.

Extracted-ion chromatogram (XIC) curves of 12 representative lipid features. Mock=uninfected; Infection=MERS-CoV infected sample. ‘_’ represent the retention time of target peak in XIC.

Q3: There was no mention of any internal standards used for LC-MS.

Response: We have used deuterated lipids including Arachidonic acid-d8, 15(S)-HETE-d8, Leukotriene-B4-d4 and Lyso-Platelet-activating Factor C16-d4 (PAF C-16-d4) as internal standards for the present study. Known concentrations of these internal standards were added to the lipid extraction solution for monitoring preparation variations as shown in the Table below. Inclusion of internal standards has also been indicated in the methodology of the manuscript (page 22 line 23).

Internal standards preparation protocol to show the volume, original and final concentrations and MS monitor mode of each internal standard spiked into 100ml of lipid extraction solution

Internal standards	Volume (μ l)	Original concentration	Final concentration	Monitored mode
AA-d8	15	2mM	2 μ M	negative
PAF-C16-d4	75	40 μ M	0.2 μ M	positive
15(S)-HETE-d8	75	20 μ M	0.1 μ M	negative
Leukotriene-B4-d4	75	20 μ M	0.1 μ M	negative

Q4: How are the lipids quantified to ascertain the changes upon infection?

Response: The lipidomics result shown in Heatmap (Fig.1b) in the present manuscript was based on 'relative' ion count intensity comparison and was generated with comparative lipidomics experiments. Thus it should be reliable as long as the sample preparation and data analysis don't have significant system variations.

In the current comparative lipidomics study, we have used the blank samples, internal standards, DNA quantification and quality control (QC) samples to perform parallel lipids analysis LC-MS experiments to monitor and to calibrate analytical/ instrumental data variations and quality. The blank samples were cell free samples subjected to the same preparation protocol, which aimed to remove false positive features and noise signals from the mass feature list. The internal standards were used to monitor lipids extraction variations in different samples. DNA concentration quantification was performed on each sample to normalize cell count variations. The QC samples are pooled samples from true samples that were analyzed in the same batch queue in between the true samples to ensure the MS signals were stable throughout the analytical batch. In addition, the CV values of all mass features in QC samples were calculated for evaluating the stability during the analytical batch. The results indicated that 76.41% of all features measured in negative mode and 73.75% of all features measured in positive mode were having CV values lower than 40%, which could reveal that our lipids analytical batch was stable with acceptable variations.

Q5: The lipids identified in Figure 1b do not conform to any standard nomenclature nor was there any key in Supplemental data to identify the lipids.

Response: We apologize for having used the improper lipids nomenclature shown in the original Fig.1b, which was generated by the Heatmap tool of MetaboAnalyst. We have now updated the lipids name in Figure 1b based on the LipidMaps nomenclature (<http://www.lipidmaps.org/>). The detailed information, lipid names and annotations were also shown in Supplementary Table 2.

Comment 2

If AM580 was able to inhibit virus replication, does the treatment of virus infected cells with AM580 restore lipidomic profile? If not, the link between AM580's ability to inhibit virus replication and lipidomic profile changes are circumstantial at best.

Response: Many thanks for the reviewer's comment. AM580 was clearly shown in our study to inhibit virus replication with a 4 log reduction of virus plaque forming units and viral load, and decrease in nucleoprotein expression by western blot and flow cytometry. This virus inhibitory effect can also be shown for many other viruses in the multicycle replication assay with an effective concentration 50% (EC₅₀) at μ M levels and with a high selectivity index at an average of 100 (Fig. 2 and Fig. 3). Furthermore, AM580 can cure mice infected by lethal doses of MERS coronavirus and influenza A H7N9 virus in mouse models. As shown in Supplementary Table 2, 68 lipids were significantly changed (including up- and down- regulations) by MERS-CoV infection and the changes in 64 out of these 68 lipids were restored to various extents after the treatment with AM580. Thus the link between AM580's ability to inhibit virus replication and the associated lipidomic profile changes are unlikely to be circumstantial.

Comment 3

In the data presented on the screening of commercial lipid library to inhibit virus replication and enhance cell viability, there are at least two other compounds that are either equal or better than AM580. What are these lipids and do they also bind to SREBPs to inhibit virus replication?

Response: Thanks to the reviewer for the insightful observation. The first compound, 25-Hydroxyvitamin D3 showed almost equally-good protection as AM580 against MERS-CoV infection in cell culture (the black dot in Fig.1d), while the second compound, aryl hydrocarbon receptor agonist FICZ, exhibited anti-H1N1 activity. As we have already mentioned in the manuscript (page 6 lines 16-17), we select AM580 for further study because of its broad-spectrum antiviral activity against both MERS coronavirus and influenza virus. Both 25-Hydroxyvitamin D3 and the aryl hydrocarbon receptor agonist FICZ do not exhibit a broad spectrum antiviral activity.

To address the reviewer's question on whether these two lipids inhibit SREBPs activity, we have evaluated the n-SREBP1 and n-SREBP2 binding activity in the presence of 25-Hydroxyvitamin D3 and FICZ, respectively. As shown below, the other two lipids are not likely to bind to SREBPs and inhibit virus replication while AM580 showed significant inhibitory effect (**p<0.01). The result has been included as the Supplementary Fig.7m.

Comment 4

Despite the liberal use of the term ‘bioactive lipid’ by the authors, AM580 is not an endogenous, naturally occurring bioactive lipid. It is a synthetic agonist of RAR α . Presumably, there could be other compounds in the ‘lipid library’ that are synthetic and calling them ‘bioactive lipids’ is misleading or a lack of understanding of what a bioactive lipid is.

Response: We agree with the reviewer and have now removed all the descriptions of AM580 as a ‘bioactive lipid’ throughout the manuscript. However, we still have to keep the name ‘Bioactive-active II lipid screening library’ un-changed (page 26, line 4) as it is the commercial name <https://www.caymanchem.com/product/10507> of the manufacturer so as to guarantee an accurate description of the materials that were used in the study.

Reviewer #4 (Remarks to the Author):

1. The manuscript by Yuan et al. describes the identification of sterol regulatory element binding protein (SREBP) as a key regulator for lipid biosynthesis and a potential cellular target for antiviral therapeutics. The authors used transcriptomics and lipidomic profiling as well as multiple molecular and biochemical methods to show that proteolytic processes and lipid biosynthesis pathways, including palmitoylation and double-membrane vesicles formation, are affected by SREBP. The latter are known to be essential for virus replication and particle formation. The authors found that a previously established compound (AM580) blocked SREBP binding to sterol regulatory elements (SREs). AM580 treatment post infection inhibited virus replication in vitro (MERS-CoV, SARS-CoV, H1N1, EV-A71, ZIKV and AvV5) and in vivo (mouse model for MERS-CoV and Influenza virus). The authors therefore claim to have identified a broad-spectrum antiviral drug.

The manuscript is overall concise and well written. The data generally support the in vitro and in vivo data. The identification of a key regulator of lipid biosynthesis is of interest, in particular, as viruses exploit this pathway to sequester by-products of virus replication (dsRNA) from immune recognition and to produce virus progeny. However, the claim to have identified a broad-acting antiviral therapeutic is in my opinion exaggerated. Neither the applied organoid intestinal model nor the relatively artificial transgenic mouse model justifies the title of the paper, i.e. “Targeting the SREBP-dependent lipidomic reprogramming by virus infection for broad-spectrum therapeutic intervention”. Importantly, off-target and other downstream effects resulting from AM580 treatment should be investigated. In particular, as the AM580-induced lack of DMV formation may affect innate immune regulation. Supplementary Figure 6 is not sufficient to exclude that AM580 affects immune regulation in context of MERS-CoV replication.

Response: Thanks for the insightful comments of the reviewer. We agree with the reviewer and have toned-down the title as ‘SREBP dependent lipidomic reprogramming as a potential broad-spectrum antiviral target’.

Additional experiments (see response #3 below) showed that AM580 did not trigger the IFN- α/β response after its addition to the MERS-CoV infected cells.

Major

2. In all Figures the authors should provide PFU/ml and virus genome equivalents/ml either in the main Figures or at least as supplemental information. As AM580 blocks DMV formation one would expect an increase of defective interfering particles.

Response: we have now provided both PFU/ml and viral copy/ml accordingly. For consistency and clarity, results in PFU/ml were shown in the main Figures, while their corresponding virus genome equivalents/ml were shown in the Supplementary Fig.7a to 7l. Note that virus titration by both plaque assay and RT-qPCR method were applied to the culture supernatant samples, while cell lysate samples were examined by RT-qPCR only.

As shown below, once AM580 was added, there is a more significant reduction in plaque forming units (***) than viral load (*) especially when Vero cell was used (about 3.5 log₁₀ reduction as measured by plaque assay VS about 2.5 log₁₀ reduction as measured by RT-qPCR). This finding is in line with the reviewer’s comment of increase of defective viral particles after treatment by AM580 and is now included in the Fig. 2d and Supplementary Fig.7b (supplementary information, page 8).

3. The compound AM580 inhibits DMV formation which likely results in increased exposure of MERS-CoV-derived dsRNA to cellular helicases with subsequent activation of the interferon pathway. Please included experiments using virus infection and AM580 treatment followed by the assessment of immune activation by, for example, real time PCR or IFN ELISA.

Response: To investigate this possibility, we studied the expression of IFN- α , IFN- β and IFN- γ in MERS-CoV-infected human monocyte-derived macrophages (MDMs), by both qPCR and ELISA methods and in the presence or absence of AM580. We chose MDMs because they are the key cells involved in the innate immune response. The qPCR result, after normalized by the mock-infected group, suggested that addition of AM580 to the virus-infected cells did not lead to an increase of antiviral IFN- β at all time-points post-infection but a decrease of IFN- α and the pro-inflammatory IFN- γ at the later time-points like 6hpi and 24hpi. By ELISA, however, secretion IFN- α and IFN- β were below the detection limit at all time-points (all b.d.) except that of IFN- γ at 6 and 24hpi.

The observation in ELISA could be explained by the antagonism of the early antiviral IFN- α/β response by MERS-CoV in our (Zhou et al., 2014) and other's (Zielecki et al., 2013) previous publications, probably mediated through viral structural, accessory, and nonstructural proteins M, ORF4a, ORF4b, ORF5, and papain-like protease (Yang et al., 2015). Another report suggests that pro-inflammatory

effects of IFN- γ is believe to outweigh its antiviral effects in SARS patient (Huang et al., 2005), thus the reduced IFN- γ production, as detected at 6 and 24hpi, were likely due to the decreased MERS-CoV replication by AM580. Therefore, we believe the inhibition of DMV formation by AM580 and the decreased shielding of dsRNA by DMV does not lead to an increased innate immune response. We have added these figures in the new Supplementary Figure 6e and revised the legends (Supplementary information, pages 6-7) accordingly.

4. Figure 2a/ Figure 2a legend & Page 7 line 3: How did the authors guarantee that the inhibitor or its downstream effects were not disturbing the PFU formation in the plaque assays? Increased IFN secretion may, for example, inhibit plaque formation as MERS-CoV is highly IFN sensitive. Please always indicate in the Figure legends when AM580 was applied to the cells. Which cells were used in Figure 2a exactly?

Response: we agree with the reviewer that there might be some residual AM580 or other factors in the infectious cell culture supernatant that may disturb the PFU formation. When applying to plaque assay, however, the dilution effect (in our case 50-fold is the lowest dilution factor and then 500, 5000, 5000-fold further) would significantly diminish these interferences to the plaque assay. As also advised by the reviewer, RT-qPCR assay was used to confirm the antiviral activity as shown in supplementary Fig.7a and also for other plaque assays in parallel.

Because there is no significant increase of IFN- α/β secretion in MERS-CoV infected cells after AM580 treatment (see response #3), this 'downstream effect should not disturb the plaque formation either.

In addition, we have checked thoroughly and made sure all the essential details (i.e. time-points, cell line and MOI) were provided in the legends. In Fig.2a, Huh7 cells were used because they are highly permissive for MERS-CoV infection and also highly active in lipid metabolism. AM580 was added after virus entry/internalization, i.e. 1h after virus-cells incubation, the infectious inoculum was aspirated, washed and replaced by fresh medium containing AM580. Please refer to the legends (page 38 lines 26-27) for detail.

5. Figure 2f-g: Intestinal organoids are an interesting model but MERS-CoV is obviously a respiratory virus. Human airway epithelial cells would be the appropriate model in this case.

Response: We agree with the reviewer and have added additional data of AM580 in term of its anti-MERS-CoV activity on human airway epithelial cells. We found that human primary small airway epithelial cells (ATCC® PCS-301-010™) were permissive to MERS-CoV infection. Indeed, AM580

exhibited substantial reduction of virus replication, as detected both intracellular and in the cell culture supernatant.

Please refer to new Fig. 2e and legends (page 38 lines 39-42) for detail. We also mentioned this primary cells in methodology (page 20, lines 9-11).

Minor

6. Figure 1: Please resize, the figure is hard to read.

Response: We have made the adjustment accordingly. Please refer to new Fig.1.

7. Figure 1 and suppl. Figure 1: Please mention the MOI and conditions in the Fig. legend

Response: We have made the revision accordingly in Figure 1 and suppl. Figure 1 legends. Please refer to page 38 lines 2-4 of main-text and page 1 of the Supplementary Information.

8. Figure 1d and Suppl Figure 1b: Please include a reference for the use of Remdesivir as positive control for MERS-CoV.

Response: We have added the reference accordingly. Please refer to Ref# 57 and page 26 line 17.

9. Figure 1c: Why does the steroid pathway not show up here, while it showed up in the KEGG enrichment?

Response: The pathway enrichment and topology results were derived from the integrated transcriptome and lipidomic analysis by MetaboAnalyst webserver targeting for metabolomics data analysis, while the KEGG pathway enrichment was conducted on transcriptome data by R package ClusterProfiler. The data source, methodology and background databases are different between these two analyses, which may lead to the difference in results.

10. Suppl Fig. 1a: Only a few genes are enriched here (10 or less), whilst GO enrichment showed 94 enriched genes. Is 10/94 reflective for the proportion of steroid biosynthesis/total lipid biosynthesis?

Response: We agree with the reviewer that the 94 genes in GO enrichment represent that there are 94 differentially expressed genes (DEGs) included in the GO term ‘lipid metabolism’; while around 10 genes are compiled in the steroid biosynthesis pathway in KEGG database. However, it is not reflective for the proportion of steroid biosynthesis among total lipid biosynthesis. Again, this is due to the variation of background databases among different analytical tool. At the present stage, we cannot find a universal standard matching database to unify GO and KEGG and other analysis tools. Therefore, it is possible that some steroid biosynthesis genes are included in KEGG, but not included in GO (vice versa), which leads to the discrepant results after analysis by these two analytical tools.

11. Figure 2b: The time-points of the N protein detection are not explained. The kinetic in Fig 2a shows the time-points beyond 12 hours until 48 hours, the WB shows 0-9 hours post infection.

Response: Because Huh7 cells are highly permissive to MERS-CoV infection, thus different time-points coupled with different multiplicity of infection (MOI) were applied. In Fig.2a, we used low MOI (=0.01) to infect Huh7 cells to evaluate the antiviral effects in a relatively longer time interval (48hpi). In Fig.2b, high MOI (=1) was applied to determine the viral N-protein expression. According to our observation, Huh7 cells will start to develop cytopathic effects (CPE) after MERS-CoV infection for 12 hours at high MOI. Therefore, we select 9hpi as the end-point to demonstrate that there was similar level of host cell β -actin but significantly less amount of viral NP protein. We have specified these conditions in the result description (page 8 line 8) and legends (page 38 lines 31-32).

12. Figure 2c: The detection of N protein by IF is not convincing in the DMSO control panel. Please improve the Figure quality.

Response: We agree with the reviewer and have repeated the staining. In the new Figure 2c, the upper panel flow cytometry data generally reflects the number of virus-infected cells, while the lower panel is an illustration of amount of viral antigen expression of individual cells.

13. Figure 3c: Please also provide information regarding the PFU/ml as mentioned above.

Response: Besides Figure 3c (MERS-CoV mouse model), we have also provided both viral load and viral titers for Fig.3g (Influenza A H7N9 mouse model). Generally, the plaque assay and qPCR results exhibited similar pattern of virus inhibition *in vivo*. As mentioned in response #1, the previous version qPCR results of mouse samples have been moved to supplementary Fig.7e and 7f, respectively.

14. Figure 6e: moi or time-point not provided in the Figure legend

Response: We have specified it accordingly, please refer to page 40 lines 40-42.

Reference

- Burnum-Johnson, K. E., Kyle, J. E., Eisfeld, A. J., Casey, C. P., Stratton, K. G., Gonzalez, J. F., . . . Metz, T. O. (2017). MPLEx: a method for simultaneous pathogen inactivation and extraction of samples for multi-omics profiling. *Analyst*, *142*(3), 442-448. doi:10.1039/c6an02486f
- Chong, J., Soufan, O., Li, C., Caraus, I., Li, S., Bourque, G., . . . Xia, J. (2018). MetaboAnalyst 4.0: towards more transparent and integrative metabolomics analysis. *Nucleic Acids Res*, *46*(W1), W486-w494. doi:10.1093/nar/gky310
- Heaton, N. S., Perera, R., Berger, K. L., Khadka, S., Lacount, D. J., Kuhn, R. J., & Randall, G. (2010). Dengue virus nonstructural protein 3 redistributes fatty acid synthase to sites of viral replication and increases cellular fatty acid synthesis. *Proc Natl Acad Sci U S A*, *107*(40), 17345-17350. doi:10.1073/pnas.1010811107
- Horton, J. D., Goldstein, J. L., & Brown, M. S. (2002). SREBPs: activators of the complete program of cholesterol and fatty acid synthesis in the liver. *J Clin Invest*, *109*(9), 1125-1131. doi:10.1172/JCI15593
- Huang, K. J., Su, I. J., Theron, M., Wu, Y. C., Lai, S. K., Liu, C. C., & Lei, H. Y. (2005). An interferon-gamma-related cytokine storm in SARS patients. *J Med Virol*, *75*(2), 185-194. doi:10.1002/jmv.20255
- Ivanova, P. T., Milne, S. B., Byrne, M. O., Xiang, Y., & Brown, H. A. (2007). Glycerophospholipid Identification and Quantitation by Electrospray Ionization Mass Spectrometry. In *Methods in Enzymology* (Vol. 432, pp. 21-57): Academic Press.
- Kaur, G., & Dufour, J. M. (2012). Cell lines: Valuable tools or useless artifacts. *Spermatogenesis*, *2*(1), 1-5. doi:10.4161/spmg.19885
- Kind, T., Liu, K. H., Lee, D. Y., DeFelice, B., Meissen, J. K., & Fiehn, O. (2013). LipidBlast in silico tandem mass spectrometry database for lipid identification. *Nat Methods*, *10*(8), 755-758. doi:10.1038/nmeth.2551
- Lau, S. K., Lee, K.-C., Lo, G., Ding, V. S., Chow, W.-N., Ke, T. Y., . . . Sridhar, S. (2016). Metabolomic Profiling of Plasma from Melioidosis Patients Using UHPLC-QTOF MS Reveals Novel Biomarkers for Diagnosis. *International journal of molecular sciences*, *17*(3), 307.
- Munger, J., Bennett, B. D., Parikh, A., Feng, X.-J., McArdle, J., Rabitz, H. A., . . . Rabinowitz, J. D. (2008). Systems-level metabolic flux profiling identifies fatty acid synthesis as a target for antiviral therapy. *Nature Biotechnology*, *26*, 1179. doi:10.1038/nbt.1500
- Nakayasu, E. S., Nicora, C. D., Sims, A. C., Burnum-Johnson, K. E., Kim, Y. M., Kyle, J. E., . . . Metz, T. O. (2016). MPLEx: a Robust and Universal Protocol for Single-Sample Integrative Proteomic, Metabolomic, and Lipidomic Analyses. *Msystems*, *1*(3). doi:10.1128/mSystems.00043-16
- Oudshoorn, D., Rijs, K., Limpens, R., Groen, K., Koster, A. J., Snijder, E. J., . . . Barcena, M. (2017). Expression and Cleavage of Middle East Respiratory Syndrome Coronavirus nsp3-4 Polyprotein Induce the Formation of Double-Membrane Vesicles That Mimic Those Associated with Coronaviral RNA Replication. *MBio*, *8*(6). doi:10.1128/mBio.01658-17
- Shimano, H., Shimomura, I., Hammer, R. E., Herz, J., Goldstein, J. L., Brown, M. S., & Horton, J. D. (1997). Elevated levels of SREBP-2 and cholesterol synthesis in livers of mice homozygous for a targeted disruption of the SREBP-1 gene. *J Clin Invest*, *100*(8), 2115-2124. doi:10.1172/JCI119746
- Silva, L. P., Lorenzi, P. L., Purwaha, P., Yong, V., Hawke, D. H., & Weinstein, J. N. (2013). Measurement of DNA concentration as a normalization strategy for metabolomic data from adherent cell lines. *Anal Chem*, *85*(20), 9536-9542. doi:10.1021/ac401559v
- To, K. K., Lee, K.-C., Wong, S. S., Sze, K.-H., Ke, Y.-H., Lui, Y.-M., . . . Hung, I. F. (2016). Lipid metabolites as potential diagnostic and prognostic biomarkers for acute community acquired pneumonia. *Diagnostic Microbiology and Infectious Disease*.

- To, K. K. W., Lee, K. C., Wong, S. S. Y., Lo, K. C., Lui, Y. M., Jahan, A. S., . . . Yuen, K. Y. (2015). Lipid mediators of inflammation as novel plasma biomarkers to identify patients with bacteremia. *Journal of Infection*, 70(5), 433-444. doi:10.1016/j.jinf.2015.02.011
- Tsugawa, H., Cajka, T., Kind, T., Ma, Y., Higgins, B., Ikeda, K., . . . Arita, M. (2015). MS-DIAL: data-independent MS/MS deconvolution for comprehensive metabolome analysis. *Nat Methods*, 12(6), 523-526. doi:10.1038/nmeth.3393
- Yang, Y., Ye, F., Zhu, N., Wang, W., Deng, Y., Zhao, Z., & Tan, W. (2015). Middle East respiratory syndrome coronavirus ORF4b protein inhibits type I interferon production through both cytoplasmic and nuclear targets. *Sci Rep*, 5, 17554. doi:10.1038/srep17554
- Zhou, J., Chu, H., Li, C., Wong, B. H., Cheng, Z. S., Poon, V. K., . . . Yuen, K. Y. (2014). Active replication of Middle East respiratory syndrome coronavirus and aberrant induction of inflammatory cytokines and chemokines in human macrophages: implications for pathogenesis. *J Infect Dis*, 209(9), 1331-1342. doi:10.1093/infdis/jit504
- Zhu, Z. J., Schultz, A. W., Wang, J., Johnson, C. H., Yannone, S. M., Patti, G. J., & Siuzdak, G. (2013). Liquid chromatography quadrupole time-of-flight mass spectrometry characterization of metabolites guided by the METLIN database. *Nat Protoc*, 8(3), 451-460. doi:10.1038/nprot.2013.004
- Zielecki, F., Weber, M., Eickmann, M., Spiegelberg, L., Zaki, A. M., Matrosovich, M., . . . Weber, F. (2013). Human cell tropism and innate immune system interactions of human respiratory coronavirus EMC compared to those of severe acute respiratory syndrome coronavirus. *J Virol*, 87(9), 5300-5304. doi:10.1128/JVI.03496-12

Reviewers' Comments:

Reviewer #1:

Remarks to the Author:

The authors have adequately responded to my query with a new experiment. This experiment which involved performing a double knockdown of SREBP1 and 2 showed a significant 2 log decrease in viral replication which supports the major conclusion of this paper that their newly identified small molecule likely targets these proteins in its MOA.

I believe the paper is suitable for publication in Nature Comm.

Reviewer #2:

Remarks to the Author:

the authors have addressed my concerns

Reviewer #3:

Remarks to the Author:

The manuscript is significantly improved with the changes incorporated in response to the comments. A few minor points to note:

1. On page 7, line 16, AM580 is referred to as a retinoid derivative. This is incorrect. The structure is far from any retinoid structure. AM580 is a synthetic agonist that binds to RxRa selectively. Hence, any reference to retinoid should only be as a synthetic agonist.

2. While the details of lipidomic analysis are needed to evaluate the manuscript, the methods can be in Supplementary data and can be removed from the main manuscript. On a technical note, the internal standards used to estimate recovery are not the best choices. Lyso-PAF is not a good representative of glycerophospholipids to be extracted similarly. Moreover, the standards are to be added to the sample before adding the extraction solvent, not to the extraction solvent. The way the authors conducted the extraction, overestimates the extraction efficiency, unrealistically minimizes variation in extraction, and does not reflect the true extraction of lipids in the sample. This is a problem even though the lipid level comparison between control and experimental samples was done directly between the identified lipids.

Finally, the identification of AM580 as an antiviral compound is purely serendipitous, despite the fact that it came out of a 'lipid' library screen (a few of the compounds in that library are neither lipids nor bioactive!). The fact that AM580 was long known as a selective agonist to RxRa and not to RxR β but binds to SREBP as shown in this study clearly demonstrates its off-target effects. While removal of the references to AM580 as a bioactive lipid was necessary, the off-target effects has to be acknowledged in the manuscript because there is no guarantee that the compound does not bind to other targets that might come to light in the future.

Reviewer #4:

Remarks to the Author:

All remarks were addressed adequately. I have no further comments.

REVIEWERS' COMMENTS:

Reviewer #1 (Remarks to the Author):

The authors have adequately responded to my query with a new experiment. This experiment which involved performing a double knockdown of SREBP1 and 2 showed a significant 2 log decrease in viral replication which supports the major conclusion of this paper that their newly identified small molecule likely targets these proteins in its MOA.

I believe the paper is suitable for publication in Nature Comm.

Thank you.

Reviewer #2 (Remarks to the Author):

the authors have addressed my concerns

Thank you.

Reviewer #3 (Remarks to the Author):

The manuscript is significantly improved with the changes incorporated in response to the comments. A few minor points to note:

1. On page 7, line 16, AM580 is referred to as a retinoid derivative. This incorrect. The structure is far from any retinoid structure. AM580 is a synthetic agonist that binds to RxR α selectively. Hence, any reference to retinoid should only be as a synthetic agonist.

We agree with the reviewer that AM580 is a synthetic agonist and have changed 'a retinoid derivative' to 'a synthetic agonist' in page 7 line 20. Moreover, we do not claim the structure-activity relationship between all retinoid derivatives and the antiviral and lipogenic inhibitory activity, which has been shown by AM580.

2. While the details of lipidomic analysis are needed to evaluate the manuscript, the methods can be in Supplementary data and can be removed from the main manuscript. On a technical note, the internal standards used to estimate recovery are not the best choices. Lyso-PAF is not a good representative of glycerophospholipids to be extracted similarly. Moreover, the standards are to be added to the sample before adding the extraction solvent, not to the extraction solvent. The way the authors conducted the extraction, overestimates the extraction efficiency, unrealistically minimizes variation in extraction, and does not reflect the true extraction of lipids in the sample. This is a problem even though the lipid level comparison between control and experimental samples was done directly between the identified lipids.

Thanks for the reviewer's expert comments and we agree with the limitations of our internal standard and lipid extraction method used. Firstly, PAF C16-d4 belongs to the glycerophospholipids class and because

PAF has many functions including cellular signaling, host defense and inflammatory response, we have included PAF C16-d4 as one of our routine internal standards. PAF C16-d4 may not be a good representative lipid to be extracted as an internal control. To this end, odd carbon chain glycerophospholipid standards such as PC (12:0/13:0), PE (17:0/17:0) and PG (17:0/17:0) might be better options with the reference from LipidMaps (<http://www.lipidmaps.org/>) methods (Ivanova, Milne, Byrne, Xiang, & Brown, 2007). Secondly, we also acknowledge the limitations in our internal standard addition method and have discussed the direct addition to the biological samples before lipid extraction. Please refer to page 18 and lines 15-18.

Finally, the identification of AM580 as an antiviral compound is purely serendipitous, despite the fact that it came out of a 'lipid' library screen (a few of the compounds in that library are neither lipids nor bioactive!). The fact that AM580 was long known as a selective agonist to RxR α and not to RxR β but binds to SREBP as shown in this study clearly demonstrates its off-target effects. While removal of the references to AM580 as a bioactive lipid was necessary, the off-target effects has to be acknowledged in the manuscript because there is no guarantee that the compound does not bind to other targets that might come to light in the future.

We have removed the word 'bioactive' in page 27 and line 10 so that throughout the manuscript, there is no reference to AM580 as a bioactive lipid. We have also acknowledged the off-target effects of AM580 in page 19 lines 18-22 that a compound can be a multi-target binder due to its intrinsic structural features.

Reviewer #4 (Remarks to the Author):

All remarks were addressed adequately. I have no further comments.

Thank you.

References:

Ivanova, P. T., Milne, S. B., Byrne, M. O., Xiang, Y., & Brown, H. A. (2007). Glycerophospholipid Identification and Quantitation by Electrospray Ionization Mass Spectrometry. In *Methods in Enzymology* (Vol. 432, pp. 21-57): Academic Press.